# Principal Components for Neural Network Initialization: A Novel Approach to Explainability and Efficiency

### Abstract

Principal Component Analysis (PCA) is a commonly used tool for dimension reduction and denoising. Therefore, it is also widely used on the data prior to training a neural network. However, this approach can complicate the explanation of eXplainable Artificial Intelligence (XAI) methods for the decision of the model. In this work, we analyze the potential issues with this approach and propose Principal Components-based Initialization (***PCsInit***), a strategy to avoid this complexity by initializing the first layer of a neural network with the principal components. We will illustrate that when this first layer (which is initialized by the principal components) is frozen, PCsInit is equivalent to applying PCA to the input and then training on the principal components, while being simpler to explain. In addition, we propose two variants PCsInit-Act (to incorporate nonlinearity) and PCsInit-Sub (for a more scalable approach), and show that the proposed techniques possess desirable theoretical properties. Moreover, as will be illustrated in the experiments, such training strategies can also allow further improvement of training via backpropagation compared to training neural networks on principal components.

## 1 Introduction

Principal Component Analysis (PCA) is a widely used dimensionality reduction technique that transforms high-dimensional data into a lower-dimensional space while preserving as much variance as possible. By identifying the principal components, which are the orthogonal directions that capture the most variance in the data, PCA helps to eliminate redundancy, improve computational efficiency, and mitigate the effects of noise. It achieves this through eigenvalue decomposition of the covariance matrix or singular value decomposition (SVD) of the data matrix. In the context of neural network training, PCA offers significant advantages by decorrelating the input features, which in turn leads to a better-conditioned Hessian matrix. This is crucial because a poorly conditioned Hessian, often caused by correlated features, can result in slow convergence during gradient-based optimization, leading to inefficiencies in training (Montavon et al., 2012). By transforming the input data into a set of orthogonal principal components, PCA ensures that gradient updates during backpropagation are more stable and effective, reducing the likelihood of oscillations or excessive adjustments along certain directions (Bishop & Nasrabadi, 2006). As a result, the optimization process becomes more efficient, with faster convergence toward the optimal solution. This makes PCA a valuable preprocessing technique for reducing the computational cost and enhancing the performance of neural networks, particularly in high-dimensional settings where the curse of dimensionality and correlated features are common (Hastie et al., 2009). Consequently, incorporating PCA in the training pipeline can lead to significant improvements in both speed and accuracy for models trained on complex, high-dimensional datasets (Jolliffe & Cadima, 2016). Therefore, it is common to ***train neural networks on the principal components (PCA-NN)***.

However, this approach also has some limitations. When PCA is applied before a neural network (PCA-NN), perturbation-based XAI methods, such as LIME (Ribeiro et al., 2016), Occlusion (Zeiler & Fergus, 2014), and Feature Permutation (Breiman, 2001), are heavily impacted because they rely on modifying input features and observing their effect on predictions. This is because perturbing the input features instead of the principal components directly affects the SVD of the input or the covariance matrix of the input, and principal components are linear combinations of multiple features. Moreover, if perturbations are applied to principal components, the results are meaningful only in the

transformed PCA space. Mapping the effects back to the original features requires using the PCA loading matrix, which defines the contribution of each original feature to each principal component. By redistributing the changes in principal components proportionally to their feature weights in the loading matrix, approximate feature-level insights can be obtained. However, this process introduces inaccuracies because the dimensionality reduction step in PCA discards some variance, and the transformation is not fully invertible. Overall, ***perturbation-based XAI methods are inherently less interpretable in PCA-NN pipelines because the transformations obscure the direct relationship between input features and predictions***.

Next, consider gradient-based XAI methods. For PCA-NN, the original input data $X$ is transformed into a lower-dimensional representation $Z = W^\top(X - \mu)$, where $W$ is the PCA projection matrix and $\mu$ is the mean of the input data, before being passed to the neural network. Gradient-based visualization methods such as saliency maps (Kadir & Brady, 2001) and Grad-CAM (Selvaraju et al., 2017), which rely on backpropagating gradients from the network's output to its input, can still be applied in this setup by mapping the gradients from the PCA-transformed space $Z$ back to the original input space $X$. This mapping can be achieved using the chain rule, where $\frac{\partial \text{output}}{\partial X} = \frac{\partial \text{output}}{\partial Z} \cdot \frac{\partial Z}{\partial X}$, and for PCA, the Jacobian $\frac{\partial Z}{\partial X} = W^\top$. Consequently, gradients computed in the PCA space can be back-projected into the original input space to enable visualization. However, ***PCA may remove or distort fine-grained input features that are crucial for effective gradient-based visualization, potentially reducing the interpretability and quality of methods like saliency maps or SmoothGrad***.

Next, while feature attribution methods like SHAP (Lundberg & Lee, 2017) and LIME (Ribeiro et al., 2016) can still explain predictions in the PCA-transformed space, mapping these explanations back to the original feature space for explanation of the input requires leveraging the PCA loading matrix and comes with the limitation of approximation. Specifically, in a PCA-NN setup, SHAP can be applied directly to the PCA-transformed features $Z$. To attribute importance to the original input features $X$, the contributions to $Z$ must be back-projected to $X$ using the PCA matrix $W$. However, $W$ may not be an invertible matrix. Therefore, the explanation is not complete, but rather an approximation. A similar thing happens to LIME. In summary, these methods can attribute importance in PCA-NN systems, but ***additional steps are required to map contributions from the reduced-dimensional PCA space back to the original input space, and the quality of these attributions depends on how much information is preserved by PCA***.

These limitations motivate us to introduce ***Principal Components based Initialization (PCsInit)***, a novel initialization technique for the first layer of a neural network based on the principal components, which is based on the similarity between the multiplication in PCA and the multiplication between the neural network input and weight matrix. Putting PCA inside the neural network via initialization, as in PCsInit, allows the explanation using XAI methods to be the same as for a regular neural network without applying PCA. Also, while PCA preprocessing requires maintaining and applying the transformation matrix during inference, PCsInit incorporates this information directly into the network weights, reducing operational complexity. Next, PCsInit allows the network to adaptively refine these initialized weights during training, potentially capturing more nuanced feature relationships that static PCA transformation might miss. Most importantly, PCsInit simplifies the explainability pipeline by eliminating the need to back-project through a separate PCA transformation step. Moreover, we also introduce PCsInit-Act, which applies an activation layer after the principal components to increase the neural network's ability to recognize nonlinear patterns. Also, we introduce PCsInit-Sub, another variant of PCsInit that computes the principal components based on a subset of the input to increase the computational efficiency for large datasets. Note that these approaches are different from Neural PCA (Valpola, 2015), which refers to using neural networks to approximate PCA.

In summary, our contributions include (i) the introduction of PCsInit, a novel method that integrates PCA into neural networks via weight initialization; (ii) we propose two variants: PCsInit-Act, which enhances nonlinear pattern recognition by applying activation functions after the PCA-initialized layer, and PCsInit-Sub, which allows efficient scaling to large datasets by computing principal components on data subsets; (iii) a theoretical analysis demonstrating the desirable properties of the proposed approach regarding convergence and robustness under noisy data; and (iv) extensive experiments across multiple datasets confirming that PCsInit and its variants achieve performance comparable to or superior to PCA-NN while maintaining a more straightforward explainability framework.

## 2 Methodologies

In this section, we will introduce ***Principal Components Analysis Initialization (PCsInit)***, its two variations (PCsInit-Act and PCsInit-Sub), and relevant theoretical properties. The main idea of PCsInit can be described as follows: First, let $\mathbf{X} = [x_{ij}]$, where $i = 1, ..., n; j = 1, ..., p$, be an input data matrix of $n$ observations and $p$ features. We assume $X$ represents tabular or vector data. For high-dimensional inputs with spatial or temporal structure (e.g., images), we assume a flattened vector representation, as PCsInit is inherently designed for fully connected architectures rather than convolutional ones. Assume also that the features are centered and scaled. Then, recall that the solution of PCA can be obtained using the singular value decomposition of $\mathbf{X}$: $\mathbf{X} = \mathbf{UDW}^T$, where $\mathbf{U}$ is an $n \times p$ orthogonal matrix, $\mathbf{W}$ is a $p \times p$ orthogonal matrix, and $\mathbf{D}$ is a $p \times p$ diagonal matrix with diagonal elements $d_1 \geq d_2 \geq ... \geq d_p \geq 0$. If $r$ eigenvalues are used, the projection matrix is $\mathbf{W}_r$, which consists of the first $r$ columns of $\mathbf{W}$. Then, the dimension-reduced version of $\mathbf{X}$ is $\mathbf{XW}_r$. This multiplication has some similarity with the multiplication between the input $\boldsymbol{X}$ and the weight matrix $\mathbf{W}$ of the first layer in a layer of a neural network. This motivates us to bring the PCA inside the neural network by simply initializing the first layer of a neural network with the principal components of the input and using no activation function for the first layer. The width of the first layer is chosen so that n_components is retained or so that a desired percentage (for example, 90%, 95%, 99%) of the variance explained is retained.

Formally, the training process of PCsInit is described in Algorithm 1. In the first step, PCA is applied to the entire input dataset $\boldsymbol{X}$ to extract the top $r$ principal components, which are stored in the projection matrix $\mathbf{W}_r$. These principal components represent the directions of maximum variance in the data, allowing for an efficient lower-dimensional representation while preserving critical information. In the second step, the matrix $\mathbf{W}_r$ is used as the weight matrix for the first layer of a neural network $f$. This initialization ensures that the first layer performs a transformation aligned with the most informative features of the input data, providing a strong inductive bias that can facilitate learning. At the third step, to stabilize training and allow the deeper layers to adapt to the PCA-based initialization, the first layer is frozen, meaning its weights remain unchanged, while the remaining layers of the network are trained for $n_{\text{frozen}}$ epochs. This prevents drastic weight updates in the first layer, ensuring that the extracted principal components provide a meaningful starting point for training. At the fourth step, after the deeper layers have sufficiently adapted, the first layer is unfrozen, allowing the entire neural network to be trained jointly for an additional $n$ epochs. This fine-tuning step enables all layers, including the first one, to update their parameters in a coordinated manner, optimizing the overall network performance and leading to better feature extraction and representation learning. The purpose of the initial frozen phase and then fine-tuning is to allow the later layers to adapt to the PCA-based feature space, establishing stable higher-level representations. Once these layers have learned effective feature combinations, unfreezing the first layer enables fine-tuning of the initial PCA-based weights while maintaining the learned feature hierarchy. This two-phase approach prevents early disruption of the meaningful PCA structure while allowing eventual optimization of all parameters.

---
**Algorithm 1 PCsInit process**

---
1: Applying PCA on the whole input $\boldsymbol{X}$ to get the projection matrix $\mathbf{W}_r$ that consists of $r$ principal components,
2: Use $\mathbf{W}_r$ as the weight matrix for the first layer of the neural network $f$,
3: Freeze the first layer, and train the remaining ones for $n_{frozen}$ epochs.
4: Unfreeze the first layer and train the whole neural network for $n$ more epochs.

---

***PCsInit versus PCA prior to neural network.*** The goal of $\boldsymbol{X}\mathbf{w}_r$ is to find the principal directions (eigenvectors of the covariance matrix) that capture the most variance in the data, while for neural networks, the weight matrix $\mathbf{W}$ is learned through backpropagation to minimize a task-specific loss function. In addition, $\mathbf{w}_r$ (the principal component directions) are computed directly from the data (using eigendecomposition or SVD) and are fixed once determined. Meanwhile, PCsInit for neural networks allows $\mathbf{W}$ to be adjusted iteratively during training for possible room for improvement.

Despite the differences, note that ***for the PCsInit approach, if the weights of the first layer are frozen throughout the training process, then its performance is the same as applying PCA to the input data and then training a model on the principal components***. This is because the first layer can be

seen as the PCA-conducting process, and the subsequent layers act as the regular neural network layers. Even in this case, using PCA in the PCsInit manner with the first layer frozen during training makes it easier to explain the model. Specifically, a neural network built upon principal components makes decisions based on a transformed feature space where the most significant variations in the data are captured. Since PCA reduces dimensionality by keeping only the most informative features, the network learns patterns in terms of these principal components rather than the original raw features. The decision can be explained by analyzing which principal components contributed most to the output, mapping them back to the original features, and using techniques like SHAP or sensitivity analysis. Meanwhile, PCsInit allows using SHAP and other XAI techniques directly on the input. Therefore, ***PCsInit offers a more straightforward explanation than explaining the decision of a neural network trained on principal components***.

However, computing the principal components based on the whole input matrix can induce computational cost. Meanwhile, if the first layer is not completely frozen throughout training, it will be fine-tuned later. Therefore, it may be more computationally efficient to compute the principal components based on a subset of the input matrix instead. This motivates us to introduce ***PCsInit based on a SUBset of input data (PCsInit-Sub)*** as described in Algorithm 2, which is a slight variation of PCsInit, where the principal components are computed based on a subset of the input matrix to reduce the computational expense associated with finding the principal components.

---
**Algorithm 2 PCsInit-Sub process**

1: Applying PCA on a subset $Z$ of the input $X$ to get the projection matrix $\mathbf{W}_r$ that consists of $r$ principal components,
2: Use $\mathbf{W}_r$ as the weight matrix for the first layer of the neural network $f$,
3: Freeze the first layer, and train the remaining ones for $n_{frozen}$ epochs.
4: Unfreeze the first layer and train the whole neural network for $n$ more epochs.

---

While the first layer of PCsInit is similar to using PCA, the activation function is not applied after the first layer. Meanwhile, activation functions are known to be important in neural networks as they introduce non-linearity, allowing the model to learn complex patterns and relationships in data. In addition, they can also enable efficient gradient propagation during training. This motivates us to introduce ***PCsInit with Activation function (PCsInit-Act)***, another variation of PCsInit, which is described in Algorithm 3. It modifies PCsInit by applying an activation function after the first layer, to improve the model's ability to capture non-linear patterns within the data. On the other hand, PCsInit-Sub focuses on computational efficiency by utilizing a subset of the input data instead of the entire dataset during initialization. As will be illustrated in the experiment section, this approach provides a lightweight yet effective alternative for handling large datasets, maintaining competitive accuracy and stability while reducing computational costs. Together, these variations improve the flexibility of the PCsInit methodology, offering improved interpretability and adaptability to diverse datasets and computational constraints.

---
**Algorithm 3 PCsInit-Act process**

1: Applying PCA on the whole input $X$ to get the projection matrix $\mathbf{W}_r$ that consists of $r$ principal components,
2: Use $\mathbf{W}_r$ as the weight matrix for the first layer of the neural network $f$,
3: Add an activation function (ReLU, LeakyReLU,...) after the first layer.
4: Freeze the first layer, and train the remaining ones for $n_{frozen}$ epochs.
5: Unfreeze the first layer and train the whole neural network for $n$ more epochs.

---

**Choice of r.** Regarding $r$, the choice of $r$ affects the percentage of variance explained (which implies the amount of information retained) after PCA. Therefore, increasing $r$ will likely increase the performance of the model to a certain threshold (for example, 95% or 99% of variance explained). However, a very high $r$ may lead to retaining noise, which may decrease the performance of the model, especially for noisy data. Also, the number of retained components $r$ acts as a structural regularizer that governs the bias-variance trade-off, balancing the risk of underfitting (high bias) from excessive compression against the risk of overfitting to noise (high variance) from insufficient filtering. More details on Appendix A.3.

**Dynamics of Fine-Tuning and weight drifts from standard PCA.** Although the weight matrix $W_1$ evolves during fine-tuning and may drift from the strict orthogonal eigenvectors of the covariance

matrix, PCsInit remains a dimensionality reduction technique by virtue of its architectural constraint. By fixing the layer width to $r$ (where $r < p$), the network is physically forced to compress the input information into a lower-dimensional bottleneck. However, unlike the static projection in PCA-NN, which optimizes purely for unsupervised variance preservation, the fine-tuned PCsInit layer transitions into a *task-optimized* dimensionality reduction. This allows the network to retain the structural benefits of compression while rotating the projection subspace to capture discriminative features that are relevant to the loss function but may not necessarily align with the directions of maximum variance. Since gradient descent updates weights in directions that minimize prediction error, and random noise is typically uncorrelated with the target variable, the optimization process is unlikely to drastically distort the weights to capture noise, provided the learning rate is controlled. Therefore, PCsInit serves to bias the model against learning noise patterns early in training, a benefit that standard random initializations—which project inputs into random, potentially noise-aligned directions—cannot provide.

## 3 THEORETICAL ANALYSIS

PCsInit acts as a form of structural regularization by injecting a strong, data-driven prior into the model. The prior is the assumption that the directions of highest variance in the input data (i.e., the principal components) are the most informative for the learning task. By initializing the first layer with these components, we are essentially guiding the network to first learn from the data's dominant, low-dimensional structure. Also, for noisy data, this initialization acts as a constraint, discouraging the model from immediately fitting noise. A standard, randomly initialized network might, by chance, align some of its initial projections with spurious, noisy features. PCsInit explicitly forces the network to start its learning process from a "denoised" and structurally meaningful representation of the data. Even after this layer is unfrozen, this strong initial bias can anchor the training process in a more robust region of the parameter space, improving generalization by reducing the risk of overfitting to noise. In addition, while a rigorous analysis for deep, non-linear networks is intractable, we can build intuition by examining a simplified single-layer linear model. Without further comments, the norm used in this paper is the Euclidean norm.

### 3.1 INITIALIZATION ANALYSIS

PCA transforms the data into uncorrelated features. Therefore, the Hessian matrix (second derivative of the loss) becomes better-conditioned after PCA, making optimization more stable. While rigorous analysis for the whole neural network is challenging, we offer a simplified case scenario to illustrate the performance of PCsInit in a simplified setting:

**Theorem 3.1** *Consider a single-layer linear neural network regression model with input data $X \in \mathbb{R}^{d \times n}$ (where each column represents a data point), target labels $Y \in \mathbb{R}^{1 \times n}$, and weight vector $V \in \mathbb{R}^{d \times 1}$. Let $J(V) = \frac{1}{2}\|V^T X - Y\|^2$ be the Mean Squared Error (MSE) loss function, whose Hessian matrix with respect to $V$ is $H$. Suppose that $r$ principal components are used, $W_r$ is the matrix that consists of the selected eigenvectors of $X$, and let $Z = W_r^T X$. Then, the Hessian of $J(V)$ with respect to $V$ has a condition number $\kappa(H_r)$ that satisfies: $\kappa(H_r) \leq \kappa(H)$.*

Hence, in this case, PCsInit leads to a better-conditioned optimization problem, potentially resulting in more stable and faster convergence of gradient-based optimization algorithms compared to training with the original data. Now, recall that a function $f(x)$ is Lipschitz continuous if there exists a constant (Lipschitz constant) $L \geq 0$ such that for all $x, y$ in the domain of $f$: $\|f(x) - f(y)\| \leq L\|x - y\|$. $L$ represents the maximum rate at which the function can change. We have the following results regarding the Lipschitz constant for the first layer of PCsInit and PCsInit-Act:

**Theorem 3.2** *Let $x \in \mathbb{R}^d$ be the input vector, and let $W_r \in \mathbb{R}^{d \times r}$ be the weight matrix of the first layer, where $r \leq d$ is the number of principal components used. The columns of $W_r$ are the principal component vectors. Assuming that the bias term is zero, i.e., the output of the first layer is $h^1 = W_r^T x$. Then, the Lipschitz constant for the first layer in PCsInit is: $\sigma_{max}(W_r) = \|W_r\|$.*

**Theorem 3.3** *Consider the first layer of a neural network with the weight matrix $W_r \in \mathbb{R}^{d \times r}$ that consists of the first $r$ principal components (PCs) of the input data, forming orthonormal rows, i.e., $W_r W_r^T = I$. Following this linear layer, an element-wise Lipschitz continuous activation function $\sigma : \mathbb{R} \to \mathbb{R}$ with Lipschitz constant $L_\sigma$ is applied. Then, the Lipschitz constant $L_1$ of the first layer operation $f_1(x) = \sigma(W_r^T x)$ with respect to the $L_2$ norm is $L_\sigma$.*

Theorem 3.2 shows that the Lipschitz constant of the first layer in PCsInit is equal to the largest singular value of the weight matrix $W_r$, which is formed by the principal component vectors. This means that the maximum amount by which the output of the first layer can change for a unit change in the input is given by the largest singular value of $W_r$. Next, Theorem 3.3 shows that the Lipschitz constant of the first layer operation is simply the Lipschitz constant of the activation function used.

Now consider the Lipschitz constant of the first layer of a neural network based on the initialization method employed. PCsInit exhibits a fixed Lipschitz constant of $||W_r||$. Since $||W_r||$ is commonly orthonormal in implementation, this means the Lipschitz constant of PCsInit is a constant $||W_r|| = 1$, a characteristic it shares with orthogonal initialization (Mishkin & Matas, 2015), which initializes weight matrices to be orthogonal and thus also possesses a constant Lipschitz constant. This inherent consistency can contribute to enhanced stability within the first layer. In contrast, He and Xavier initializations result in random Lipschitz constants that differ across initializations, introducing variability that can influence training dynamics.

## 3.2 ROBUSTNESS TO NOISY DATA

Note that PCA decorrelates the input features. If the noise is independent across the original input features, then PCsInit also decorrelates the noise in the output of the first layer. Meanwhile, He and Xavier initialization can create correlations in the noise, as they involve random linear combinations of the input dimensions. Moreover, note that while scaling down the norm of any standard initializer can help denoising, it reduces the magnitude of both the signal and the noise indiscriminately. This can be detrimental, as suppressing the signal too much can lead to vanishing gradients and hinder the learning process. In contrast, PCsInit provides a principled, data-driven structure. The first layer is initialized with principal components, which are the directions of maximum variance in the data. This means that the initialization is explicitly designed to preserve the signal's most important structural information while discarding dimensions with low variance, which are more likely to be dominated by noise. Therefore, PCsInit aims to maximize the signal-to-noise ratio (SNR) in the first layer's output, whereas simply reducing the norm of a random matrix may decrease the overall magnitude but can also worsen the SNR. In addition, we present the following statements regarding the robustness of PCsInit to noisy data, which illustrates that PCA projects the input noise onto the principal subspace. The variance of the noise in each principal component direction is scaled by the corresponding eigenvalue of $X^T X$.

**Theorem 3.4** *Assume that $\tilde{x} = x + \eta$, where $\tilde{x}$ is the noisy input, $x$ is the clean input, and $\eta$ is the noise vector. In addition, assume also that $\eta \sim \mathcal{N}(0, \sigma^2 I)$, i.e., the noise follows a Gaussian distribution with zero mean and covariance matrix $\sigma^2 I$. Here, $\sigma^2 \in \mathbb{R}^+$ and $I$ is the identity matrix. Next, let the eigenvalues of $X^T X$ be $\lambda_1, \lambda_2, ..., \lambda_r$. Then, for PCsInit, the noise propagated after the first layer is $W_r^T \eta$ follows a Gaussian distribution with mean 0 and its covariance matrix is a diagonal matrix with entries $\sigma^2 \lambda_1, \sigma^2 \lambda_2, ..., \sigma^2 \lambda_r$.*

Consider other initialization schemes for the first layer: if the input is noisy ($\tilde{x} = x + \eta$), the output of the first layer is $W_1 \tilde{x} = W_1 x + W_1 \eta$. The noise component is $W_1 \eta$. So, He and Xavier initializations transform noise by multiplying it with a randomly initialized matrix with a variance scaling. Meanwhile, orthogonal initialization rotates the noise vector without changing its norm. Next, we have the following theorem regarding the bound for the noise.

**Theorem 3.5** *Consider a neural network where the first layer's weight matrix, $W_r \in \mathbb{R}^{d \times r}$, is initialized as in PCsInit, and therefore $W_r$ is orthonormal. Suppose that the input is corrupted by additive white noise, i.e., $\tilde{x} = x + \eta$, where $x \in \mathbb{R}^d$ is the clean input signal and $\eta \sim \mathcal{N}(0, \sigma^2 I)$ is the additive white noise. Then, the norm of the noise component after the first layer is preserved, i.e., $||W_r^T \eta|| = ||\eta||$. Also, let $f : \mathbb{R}^d \to \mathbb{R}^r$ be the neural network function, decomposed into layers $f = f_L \circ f_{L-1} \circ ... \circ f_1$, where $f_i$ represents the $i$-th layer, $f(x)$ is the clean output, and $\tilde{f}(\tilde{x})$ is the noisy output. Further assume that each subsequent layer $f_i$ for $i = 2, ..., L$ is $L_i$-Lipschitz continuous. Then, $||\tilde{f}(\tilde{x}) - f(x)|| \leq \left( \prod_{i=2}^{L} L_i \right) ||\eta||$.*

While the norm of the noise in the first layer is preserved for PCsInit, for other initialization methods (e.g., He, Xavier) where the first layer's weight matrix $W'$ does not have orthonormal rows, $||W'\eta||$ is not guaranteed to be equal to $||\eta||$ and can be larger or smaller. In fact, we have $||W'\eta|| \leq ||W'||||\eta||$, and it is possible that $||W'|| > 1$. So, the noise can be amplified in the first layer. Next, we have the following result, which provides deeper details regarding the noise of layer 2 and subsequent layers:

**Theorem 3.6** *For any layer $\ell > 1$, let $\rho^\ell$ is the activation function, $W^\ell \in \mathbb{R}^{d_l \times d_{l-1}}$ is the weight matrix, and $b^\ell \in \mathbb{R}^{d_l}$ is the bias vector. Then, the output at layer $\ell$ is: $h^\ell = \rho^\ell(W^\ell h^{\ell-1} + b^\ell)$. Next, let $\eta^\ell$ represents the noise in the output of layer $\ell$ and assume also that $\rho^\ell$ is Lipschitz continuous with Lipschitz constant $L_\ell$, i.e. $||h^\ell - \tilde{h}^\ell|| \le L_\ell ||W^\ell|| \cdot ||\eta^{\ell-1}||$, where $\tilde{h}^\ell$ is the clean output and $h^\ell$ is the noisy output. Then, the bound for the noise propagated to the second layer is $||\eta^2|| \le L_2 ||W^2|| \cdot ||W^1|| \cdot ||\eta^0||$. In addition, the general noise bound for any layer $\ell > 1$ is: $||\eta^\ell|| \le \left[ \prod_{i=2}^{l}(L_i ||W^i||) \right] \cdot ||W^1|| \cdot ||\eta^0||$.*

This theorem provides an upper bound on how input noise propagates through the network and shows that the noise at any layer l is directly proportional to $||W^1||$, the norm of the first layer's weight matrix. For PCsInit, ($||W^1|| = 1$). The first layer, therefore, does not amplify the input noise. Meanwhile, with He or Xavier initialization, $W^1$ is a random matrix whose norm is not strictly controlled. Depending on the dimensions and the specific random draw, its norm can be greater than 1, leading to an initial amplification of noise. By guaranteeing $||W^1|| = 1$, PCsInit prevents a potential explosion of noise magnitude at the very first layer. This is crucial because any amplification in the first layer will be propagated and potentially magnified through all subsequent layers.

## 4 EXPERIMENTS

### 4.1 EXPERIMENT SETTINGS

We conduct experiments on seven datasets with various characteristics whose details are found in Table 2. Among them, Parkinson and Micromass (Dua & Graff, 2017) have the number of features significantly higher than the number of samples, MNIST (Deng, 2012) and CIFAR-10 (Krizhevsky, 2009) are image data, and HTAD (Garcia-Ceja et al., 2021) is a noisy sensor-collected dataset. To demonstrate the effectiveness of the proposed strategies, we compare them with PCA-NN, Raw Multilayer Perceptron (Raw MLP), and Zero-phase Component Analysis MLP (ZCA MLP). For PCsInit, the first layer weights were initialized using principal components derived from the training dataset, while the first layer of the standard NN was initialized using the specified initialization technique (He, Xavier, or Orthogonal initialization). Subsequent layers in both PCsInit and the standard NN, as well as all layers of the PCA-NN (which operated on pre-computed principal components), also employed one of three initialization techniques. For PCsInit-Sub, the initial PCA was performed on a randomly selected 20% subset of the training data. For PCsInit-Act, the ReLU activation function is used for the first layer. In the PCsInit family, the first layer is initialized with the principal components, and then it is frozen for the first 30 epochs. After that, we unfreeze the first layer and train the whole model for 170 more epochs. To facilitate a fair comparison, for all strategies, all layers except the first layers of the PCsInit family are initialized with the same weights. The first layer of PCsInit and its variants are initialized based on principal components. More details are in the Appendix. The codes are released on Github.

### 4.2 RESULTS AND ANALYSIS

Due to the page limit, the results of performance loss and accuracy are reported in figures in Appendix A.2. The results illustrate the performance of the proposed techniques against PCA-NN. Notably, the PCsInit-Act variation shows strong performance on the given datasets. In addition, when dealing with noisy datasets, the PCsInit family generally performed better than PCA-NN. For instance, on the HTAD dataset, a noisy dataset, PCsInit and its variations were particularly effective. They were often the most accurate, learned quickly and steadily, and performed better than PCA-NN. Similarly, on the MNIST dataset with added Gaussian noise, the PCsInit-Act variation performed much like the other PCsInit variations. Note that PCA-NN gives the same performance as PCsInit with the first layer frozen during training. Hence, this suggests the benefits of the fine-tuning phases in PCsInit. For datasets with a large number of features compared to the sample size, such as Micromass (figure 8) and Parkinson (figure 7), the PCsInit family gives pronounced improvement compared to PCA-NN. On Micromass, while standard PCsInit still works well, its variations (PCsInit-Act and PCsInit-Sub) offer enhanced stability and competitive accuracy.

Regarding interpretability, figures 1, 2, and 3 show that for PCsInit, feature contributions can be computed directly by using the Kernel SHAP method, whereas in the PCA-NN method, they can only be approximated by first calculating SHAP values of the principal components and then deriving feature importance based on the contribution of each feature to those components. Moreover, figure 1

featuring the global feature importance for class 0 obtained using (a) PCsInit and (b) Raw MLP, shows that it is straightforward to identify the features that directly contribute to the predictions, with the top 10 features exhibiting the strongest influence. Among these, 5 features overlap between the two methods. In contrast, Figure 2 (a) shows the global principal component importance in the PCA-NN method, which highlights the contribution of principal components rather than original features. To further interpret the feature contribution, the heatmap in Figure 2 (b) illustrates the importance of each original feature through each principal component. For instance, in Class 0, features indexed by 7, 17, 23, 24, and 34 strongly influence predictions through principal component 2 (PC_2), as indicated by the more intense colors. However, determining how each feature impacts the model's predictions in this case requires more effort and further analysis.

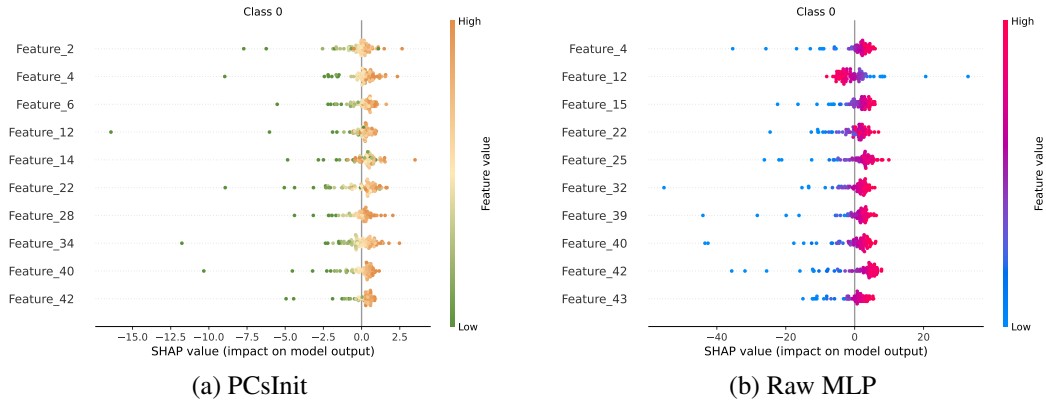

Figure 1: Top 10 most important features contributing to (a) **PCsInit** and (b) **Raw MLP** predictions on the test set of Heart dataset, Class 0. **The first layer of PCsInit is frozen throughout the training.**

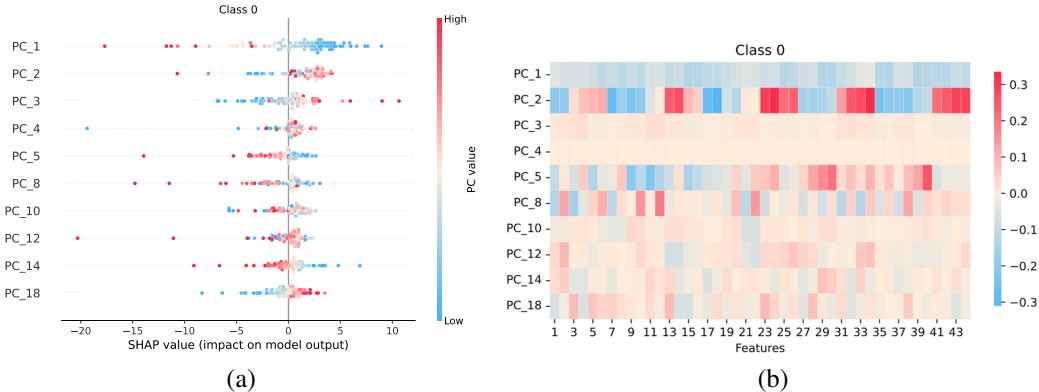

Figure 2: (a) Top 10 most principal components contributing to **PCA-NN** predictions on the Heart dataset for Class 0 across the test set. (b) Heatmap of feature importance through these components.

**Local Feature Importance.** Figure 3 (a) shows the SHAP values for a sample point using PCsInit. Here, feature_28 has the strongest influence on the outcome for both Class 0 and Class 1. Since PCsInit directly computes feature contributions without projection, it allows for identifying the driving factors behind individual predictions easily. By contrast, Figure 3 (b) illustrates the approximate SHAP values for the same sample point through PC_2 using the PCA-NN method. It should be noted that across different principal components, the same features may show different SHAP values. This makes two layers of interpretation: first, feature contributions are dispersed across multiple components; second, each component contributes to the final prediction with a different weight. As a result, the relationship between original features and the final predictions is indirect and more difficult to trace.

**Quantitative XAI metrics.** Metrics—faithfulness, stability, and fidelity—are crucial factors for ensuring the reliability and trustworthiness of explanations (Ali et al., 2023; Miró-Nicolau et al.,

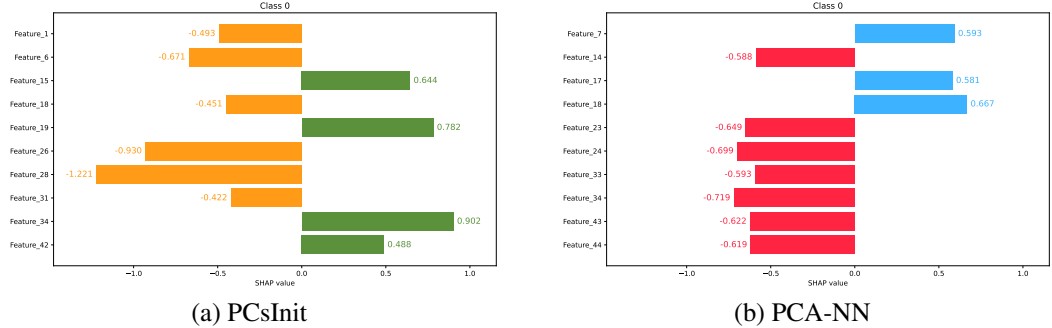

(a) PCsInit                (b) PCA-NN

Figure 3: The local feature importance of a data point contributing to the **PCsInit** prediction and **PCA-NN** prediction through $PC\_2$ on the Heart dataset for Class 0.

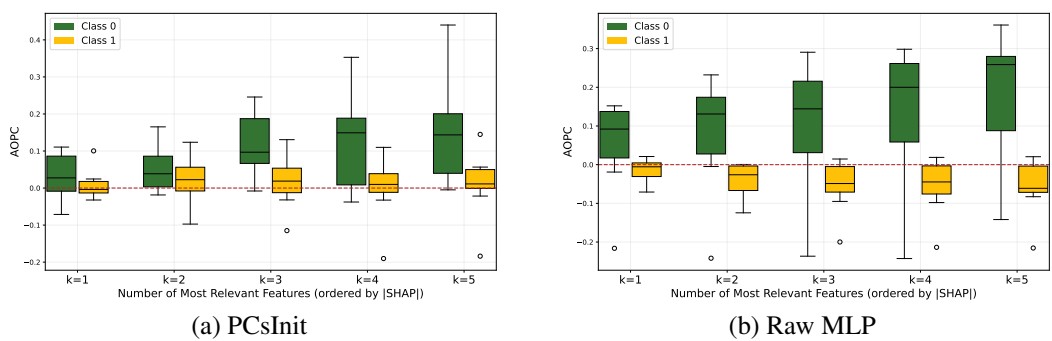

(a) PCsInit                (b) Raw MLP

Figure 4: AOPC distribution per class across 10 runs with different random seeds using zero perturbation, where the number of Most Relevant First features removed ranges from 1 to 5 on the Heart dataset under the (a) PCsInit and (b) Raw MLP.

2024; Mersha et al., 2024). Faithfulness describes the sensitivity of a model's output when the input features are perturbed, and it can be measured by Area Over the Perturbation Curve (AOPC) (Samek et al., 2017). Figure 4 presents the AOPC distributions obtained by removing the top 1 to 5 Most Relevant First (MoRF) features, ordered by their absolute SHAP values, using zero perturbation across 10 runs on the Heart dataset. Overall, while both PCsInit and Raw MLP yield positive AOPC scores for class 0, their behaviors diverge for class 1: PCsInit remains positive, whereas Raw MLP becomes negative. This indicates that the top-ranked features consistently play a meaningful role in the PCsInit's predictions for both classes. Furthermore, stability measures the consistent results of the explanations produced by an XAI method when the same input is evaluated across different model runs. It is quantified using the variance of the explanation values across runs (Mersha et al., 2024), with lower variance indicating greater stability. Across 10 runs with different random seeds, SHAP achieves mean variance scores of 0.622 and 1.130 under PCsInit and Raw MLP, respectively (Table 1), highlighting the higher stability of PCsInit explanations. Fidelity, in contrast, refers to how XAI explanations are close to the actual decision. To evaluate fidelity, we generate five synthetic binary classification datasets following the *ssin* design used in (Cortez & Embrechts, 2013), each containing 2000 samples and 20 features. The number of predefined important features that directly influence the target ranges from 3 to 7 across the five datasets, while the remaining features are irrelevant or noisy (i.e., the first dataset has 3 important features, the second has 4, and so on). We then compare the top important features identified by SHAP under the PCsInit and Raw MLP models with the ground-truth important features. Table 1 shows that the mean recovery scores of both methods are above 80%, demonstrating high fidelity in capturing the underlying ground-truth feature structure.

In general, the above analysis demonstrates that our PCsInit method provides more direct, transparent, stable, and faithful explanations, while PCA-based methods increase the complexity in model interpretability.

Table 1: Recovery scores across five synthetic datasets with different numbers of important features, mean recovery, and variance scores under the PCsInit and Raw MLP methods.

| Method | Recovery Score | | | | | | Variance of explanations |
| | #=3 | #=4 | #=5 | #=6 | #=7 | Mean | |
|---|---|---|---|---|---|---|---|
| PCsInit | 100% | 100% | 80% | 66.67% | 71.43% | **83.62%** | **0.622** |
| Raw MLP | 100% | 100% | 80% | 66.67% | 51.14% | 80.76% | 1.130 |

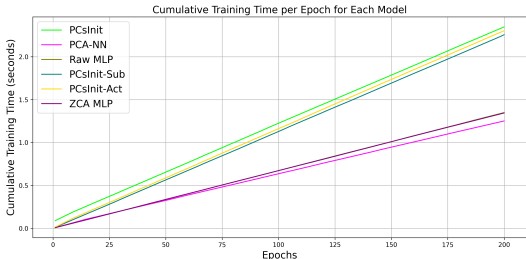

Figure 5: Running time on the Heart dataset with He initialization

**Running time.** Regarding running time, consider Figure 5, one can see that PCsInit-Sub proved to improve the running time by computing the principal components on a subset of the data. However, as the number of epochs increases, the PCsInit family has a higher cumulative running time compared to PCA-NN. This is because for PCA-NN, the input is the principal components, and therefore has fewer features. Meanwhile, for the PCsInit family, the input size is the same as the original input, as the network is initialized with PCA instead of being trained on the principal components.

## 5 CONCLUSION

In this work, we introduced PCsInit, a novel neural network initialization technique designed to embed Principal Component Analysis (PCA) within the network's first layer. A key advantage of the PCsInit family is the enhanced clarity and directness it brings to model explainability, avoiding the complexities associated with applying PCA as a preprocessing step. However, a drawback of PCsInit is that it requires computing PCA on the entire dataset, which can be computationally expensive for high-dimensional data (Though, it is also worth mentioning that PCA-NN also requires computing PCA on the entire dataset). PCsInit-Sub is a more scalable approach as it requires computing PCA only on a subset of the data, without sacrificing the accuracy, as illustrated in the experiments. Another drawback of PCsInit stems from the fact that it uses PCA, which may lead to a loss of spatial information if compared to a standard neural network without PCA (however, PCA-NN also suffers the same issue). This means that PCsInit may not be suitable for convolutional neural networks. Therefore, to mitigate this issue, we will consider dimension reduction techniques that keep spatial information, such as SpatialPCA, for example.

Moreover, our future research will delve into the broader applicability of PCsInit across different network architectures and data characteristics, including the exploration of Kernel PCA-based initialization. In addition, we will examine the performance of the proposed approaches under various neural network architectures and various types of data (noisy, imbalanced). In addition, we will explore the potential usage of Kernel PCA instead of PCA for initializing neural networks, as well as analyze in depth the choice of initialization techniques for the subsequent layers after the first layer for the PCsInit family. Also, a sparse version of PCsInit could be designed, for example, by using Sparse PCA instead of standard PCA to generate the initial weights for the first layer. This would not only reduce computational time but could also potentially improve interpretability, as each component would be defined by a smaller, less dense set of original features. Last but not least, incorporating PCsInit into the first layer of an autoencoder would be another direction, as it offers a strategic "warm-start" by initializing the model within the maximal-variance subspace, thereby enabling the network to bypass learning basic linear structures and immediately focus on capturing complex non-linear residuals.

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

## A    APPENDIX

### A.1    DATASET AND EXPERIMENTS DESCRIPTION

For all datasets except MNIST and CIFAR-10, the experiments are repeated 10 times with 70%/30% train/test split ratio, and the average is reported in the graphs. For the MNIST and CIFAR-10 datasets, the images are flattened before further processing, and we simply used the already split train-test portions.

For PCA, we choose the number of principal components so that a minimum of 95% of variance is retained, as commonly used in various research articles (Nguyen et al., 2022; Audigier et al., 2016; Do et al., 2024; Nguyen et al., 2023). The width of the network used for each dataset has a width equal to the number of principal components retained. To ensure scale invariance during the extraction of principal components, all input features were standardized prior to model initialization. Specifically, we applied normalize the data, transforming features to have zero mean and unit variance on the training set. The same transformation parameters were applied to the test set. All models are trained using a neural network architecture consisted of a 5-layer Multilayer Perceptron (MLP), with a total of 200 epochs, a batch size of 64 using Cross Entropy Loss and Adam optimizer.

Note that when the first layer is frozen, PCsInit is equivalent to PCA-NN. Therefore, the frozen case of the PCsInit is also PCA-NN in the experiments. Hence, considering the case where the first layer

of PCsInit is not frozen, then the training procedure for PCsInit involved a two-phase schedule. In the first phase, the weights of the PCA-initialized layer were frozen for $n_{frozen} = 30$ epochs. This stabilization phase allows the subsequent layers to adapt to the fixed principal component features. In the second phase, the first layer was unfrozen, and the entire network was fine-tuned for the remainder of the training duration (totaling 200 epochs).

The experiments were run on a GPU T4x2. The codes for the experiments will be made available upon acceptance.

Table 2: Descriptions of datasets used in the experiments

| Dataset | # Classes | # Features | # Samples |
|---------|-----------|------------|-----------|
| Heart | 2 | 44 | 267 |
| Ionosphere | 2 | 32 | 351 |
| Parkinson | 2 | 754 | 756 |
| Micromass | 10 | 1087 | 360 |
| HTAD | 7 | 54 | 1386 |
| MNIST | 10 | 784 | 60000 |
| CIFAR10 | 10 | 3072 | 60000 |

## A.2 EXPERIMENT RESULTS AND FURTHER DISCUSSION

In the PCsInit-Sub variant, subsets are generated by randomly sampling a fixed percentage (e.g., 20%) of the training instances to compute the initial principal components. This strategy yields a favorable accuracy-to-compute trade-off, as it significantly reduces the computational overhead of the initial eigendecomposition while maintaining stability and accuracy comparable to the full-data initialization. Unlike streaming or incremental PCA, which continuously updates the covariance model to accommodate new data, PCsInit-Sub relies on a static, one-time approximation and, therefore, is likely to be less computationally expensive. However, this subset-based approach risks "breaking down" in scenarios where critical features are rare or sparse, as a random sample may fail to capture the variance of these infrequent but discriminative signals, which are inherently discarded or undersampled compared to high-variance global structures.

Next, the model's sensitivity to the first-layer freeze duration represents a critical trade-off between preserving the robust, noise-filtering structure of the principal components and allowing the network to adapt to the specific supervised task. If the freeze duration is too short, the immediate backpropagation of large error gradients from the randomly initialized upper layers risks destroying the orthonormal PCA subspace before the network can utilize its stability, effectively negating the intended noise-reduction benefits. Conversely, extending the freeze duration indefinitely constrains the model to the static PCA subspace, effectively reverting its performance to that of a standard PCA-NN and preventing the first layer from refining its features to capture discriminative signals that may not align with maximum variance. Consequently, the results are sensitive to this parameter because it defines the necessary transition point where the model shifts from a rigid, unsupervised prior to a flexible, task-optimized feature extractor.

The experimental results depicted in the figures demonstrate that PCsInit variations outperform the ZCA MLP initialization across most datasets and initialization techniques. Additionally, our results indicate a similar performance to that of PCA-NN and Raw MLP, but are more stable than Raw MLP, as evidenced by the confidence interval. Meanwhile, as shown in the main paper, the PCsInit family is simpler and more straightforward in terms of explainability.

Next, note that when the first layer of PCsInit is not frozen, then the training procedure for PCsInit involves a two-phase schedule. In the first phase, the weights of the PCA-initialized layer were frozen for a specified number of epochs (in the experiments, $n_{frozen} = 30$ epochs). This stabilization phase allows the subsequent layers to adapt to the fixed principal component features. In the second phase, the first layer was unfrozen, and the entire network was fine-tuned for the remainder of the training duration. Therefore, even though the paper chooses the default $n_{frozen} = 30$ epochs, it is also reasonable to leave the first layers frozen, and then train the other layers until the loss saturates;

after that, unfreeze the first layer and fine-tune the whole network, as in a standard transfer learning process.

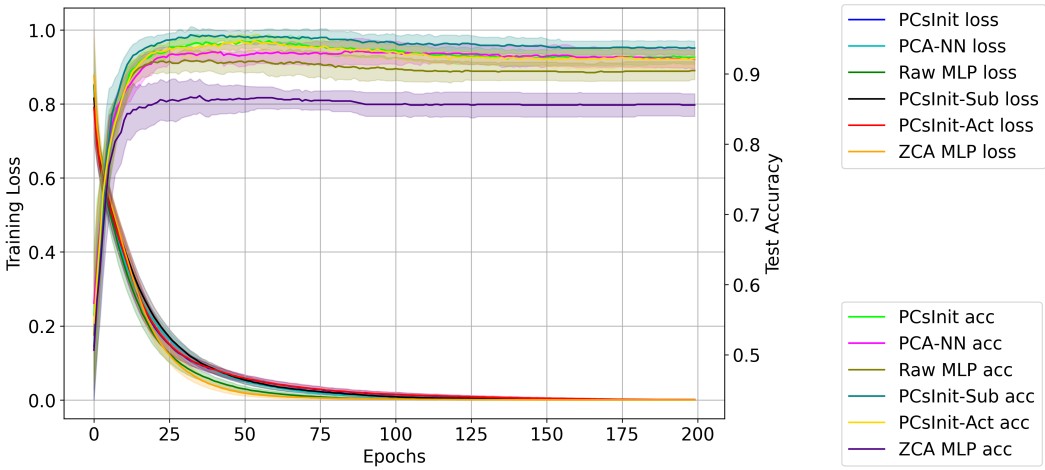

Figure 6: Ionosphere dataset with He initialization and uncertainty

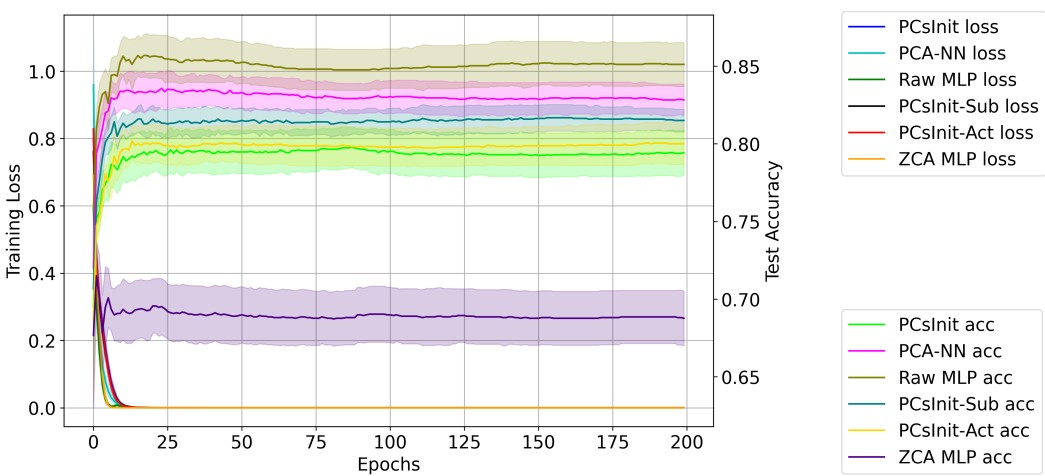

Figure 7: Parkinson dataset with He initialization and uncertainty

## A.3   MORE ON BIAS AND VARIANCE TRADE OFF

For PCsInit and its variants, the number of retained principal components, $r$, serves as a crucial structural hyperparameter that directly governs the model's position on the bias-variance tradeoff curve. When $r$ is set to a small value, the first layer acts as a narrow information bottleneck, aggressively filtering out input dimensions with lower eigenvalues. This regime typically results in high bias and low variance: the model is stable and robust to noise because it ignores the idiosyncratic fluctuations of the data, but it risks underfitting if the target variable relies on subtle, fine-grained features that were discarded during the dimension reduction.

Conversely, selecting a large value for $r$ (approaching the total number of features) preserves nearly all the information present in the input, shifting the model towards a low bias and high variance state. In this regime, the initialization captures complex relationships and fine details, minimizing systematic error. However, because the lower-variance components often contain a lower Signal-to-Noise Ratio (SNR), a large $r$ allows the network to ingest and learn from random noise patterns, making the model more susceptible to overfitting and less consistent when generalizing to unseen data.

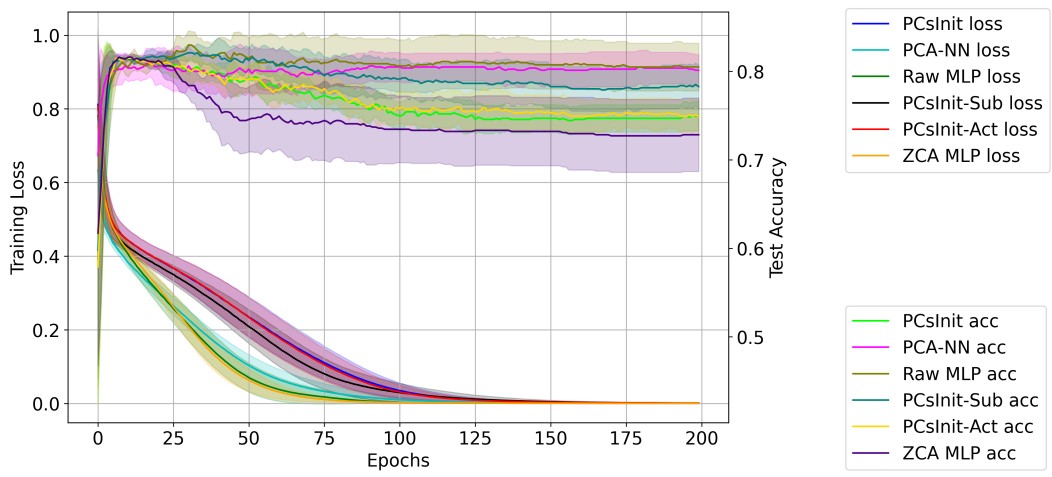

Figure 8: Micromass dataset with He initialization and uncertainty

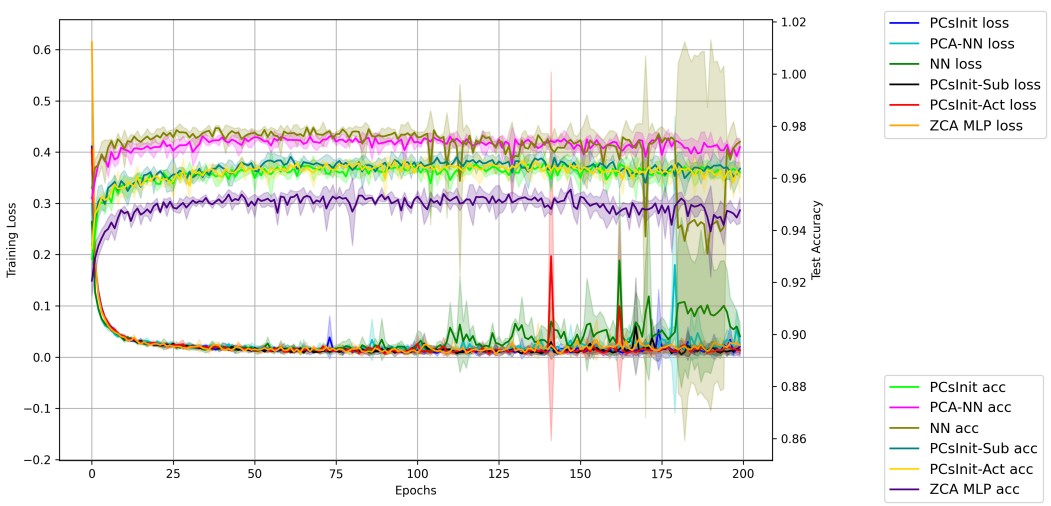

Figure 9: MNIST dataset with He initialization and uncertainty

### A.4 ATTRIBUTION DRIFT

To quantify how much the weights depart from the original PCA subspace when PCsInit has a fine-tuning phase, we calculate the subspace similarity—a metric for measuring the distance between subspaces (Jain et al., 2012)—between the initial $W_r$ matrix and the final, fine-tuned $W_1^{optimal}$ matrix by using principal angles across several datasets. Table 3 reports the averaged principal angles in degrees, where smaller angles indicate greater alignment between subspaces. Small angles for the Heart and Ionosphere datasets are in turn $6.2°$ and $2.9°$, demonstrating the first-layer weights hardly change subspace during training. This means that the PCA initialization captures the main structure of the data. However, larger angles for Parkinson ($31.9°$) and Micromass ($36.8°$) imply that the model can change its direction to optimize performance.

### A.5 PROOF OF THEOREM 3.1

**Statement.** Consider a single-layer linear neural network regression model with centered input data $X \in \mathbb{R}^{d \times n}$ (where each column represents a data point), target labels $Y \in \mathbb{R}^{1 \times n}$, and weight vector $V \in \mathbb{R}^{d \times 1}$. Let $J(W) = \frac{1}{2}\|V^T X - Y\|^2$ be the Mean Squared Error (MSE) loss function, whose Hessian matrix with respect to $V$ is $H$. Suppose that $r$ principal components are used, $W_r$ is the

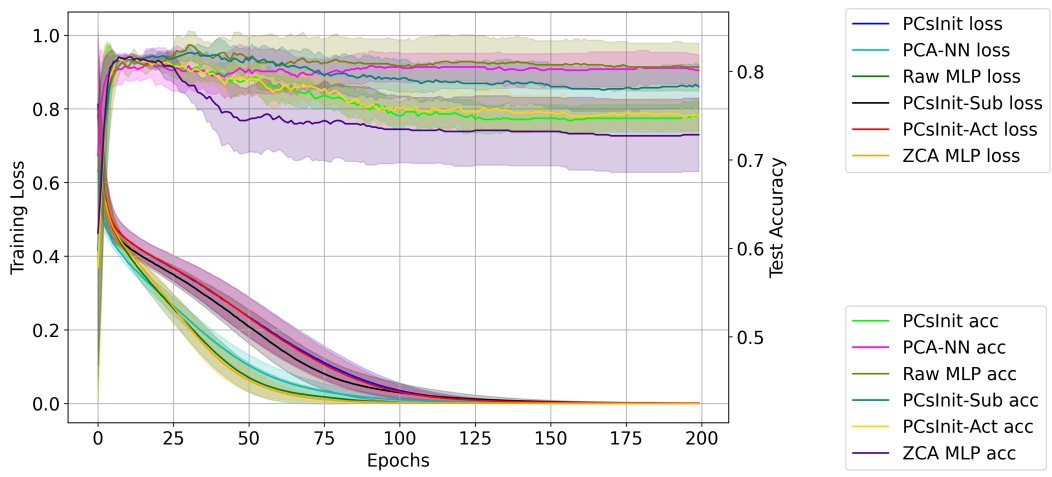

Figure 10: Heart dataset with He initialization and uncertainty

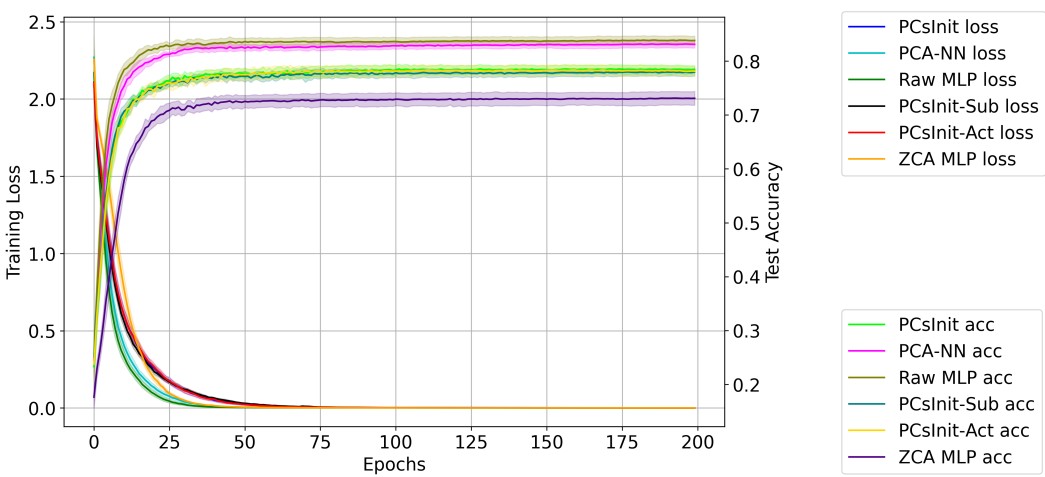

Figure 11: HTAD dataset with He initialization and uncertainty

matrix that consists of the selected eigenvectors of $X$, and let $Z = W_r^T X$. Then, the Hessian of $J(V)$ with respect to $V$ has a condition number $\kappa(H_r)$ that satisfies:

$$\kappa(H_r) \leq \kappa(H). \tag{1}$$

**Proof A.1** *It is established that the steepest descent method's convergence rate is significantly impacted by the condition number of the Hessian matrix and that convergence can be significantly slow when the condition number of the Hessian matrix is large (Ackleh et al., 2009). For a single-layer linear regression problem with the output defined as $\hat{Y} = V^T X$, the Mean Squared Error (MSE) loss function is given by:*

$$J(V) = \frac{1}{2}\|V^T X - Y\|^2.$$

*The gradient of the loss function with respect to $W$ is:*

$$\nabla J(V) = X(V^T X - Y)^T = X(X^T V - Y^T).$$

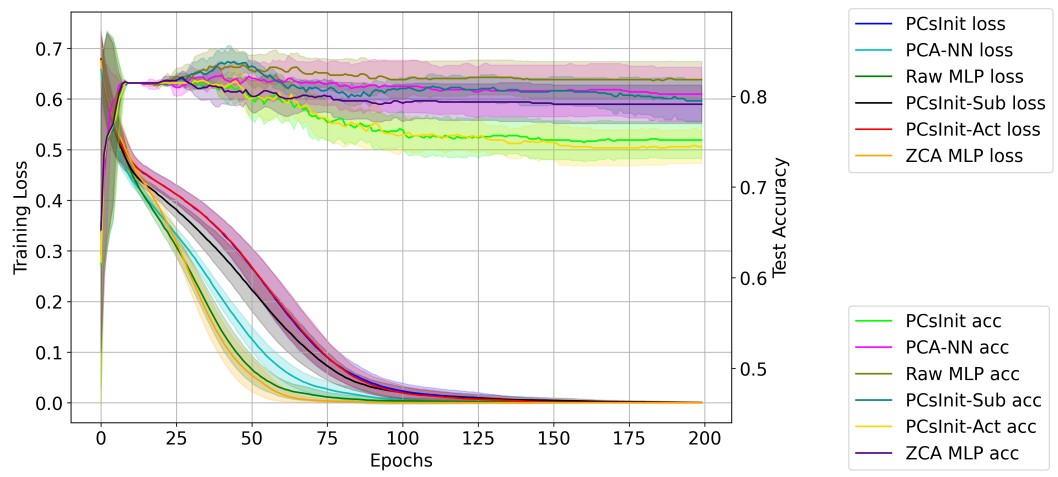

Figure 12: Micromass dataset with Xavier initialization and uncertainty

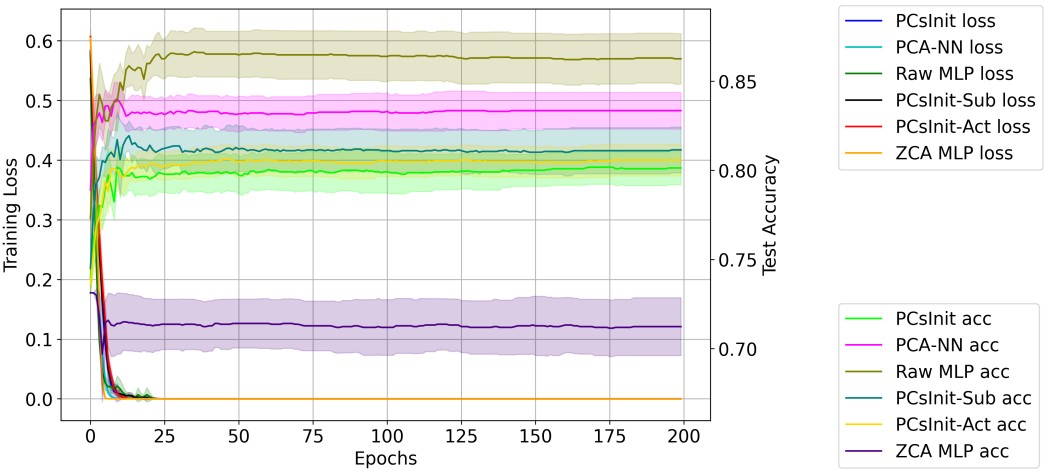

Figure 13: Parkinson dataset with Xavier initialization and uncertainty

*Therefore, the Hessian matrix is:*

$$H = \nabla^2 J(V) = XX^T. \tag{2}$$

*The condition number of the Hessian matrix remains defined as the ratio of its largest eigenvalue ($\lambda_{max}$) to its smallest positive eigenvalue ($\lambda_{min}$):*

$$\kappa(H) = \frac{\lambda_{max}}{\lambda_{min}}.$$

*Since $W_r \in \mathbb{R}^{d \times r}$ be the matrix formed by the first $r$ eigenvectors of $XX^T$, the projected data is $Z = W_r^T X \in \mathbb{R}^{r \times n}$. Therefore, the prediction is $W_r^T Z = W_r^T W_r^T X$. Based on 2, we see that the Hessian of the loss function with respect to $W_r$ in the reduced space would be $H_r = ZZ^T = (W_r^T X)(W_r^T X)^T = W_r^T XX^T W_r$.*

*Let $U \in \mathbb{R}^{d \times d}$ be the semi-orthonormal matrix whose columns are the eigenvectors of $XX^T$, ordered by decreasing eigenvalue. Since $U$ diagonalizes $XX^T$ such that $U^T XX^T U = \Lambda =$*

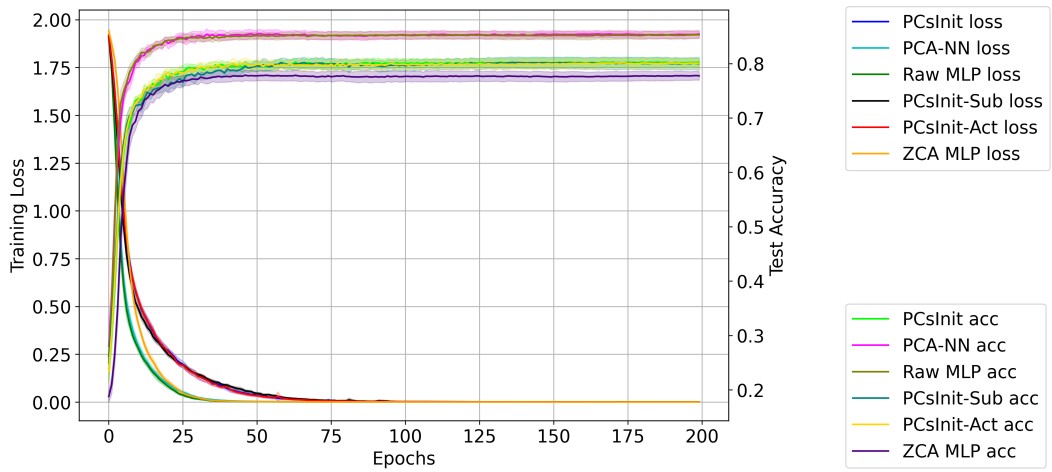

Figure 14: HTAD dataset with Xavier initialization and uncertainty

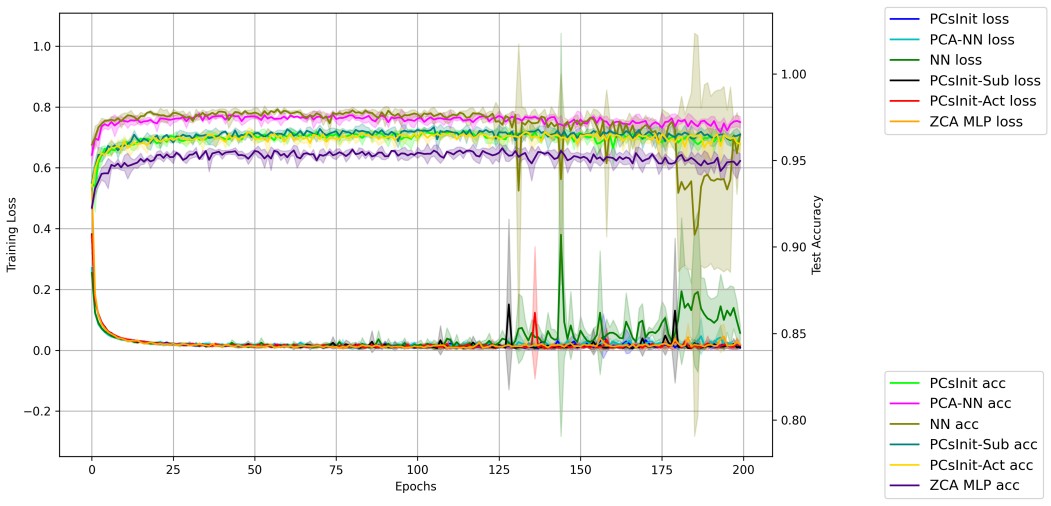

Figure 15: MNIST dataset with Xavier initialization and uncertainty

$diag(\lambda_1, \lambda_2, ..., \lambda_d)$ with $\lambda_1 \geq \lambda_2 \geq ... \geq \lambda_d \geq 0$, then $W_r^T X X^T W_r = \Lambda_r = diag(\lambda_1, \lambda_2, ..., \lambda_r)$. By discarding the smallest eigenvalues (corresponding to directions of low variance), PCA effectively works with a Hessian $H_r$ whose condition number is:

$$\kappa(H_r) = \frac{\lambda_1}{\lambda_r}$$

It is known that the steepest descent method for nonlinear optimization is highly sensitive to scaling and the problem's condition number, and that convergence can be significantly slow when the condition number of the Hessian matrix is large Ackleh et al. (2009). Since $\lambda_r \geq \lambda_{min}$ (where $\lambda_{min}$ is the smallest positive eigenvalue of the original $H = XX^T$), we have $\kappa(H_r) \leq \kappa(H)$. Furthermore, if some of the smallest eigenvalues of $H$ are close to zero, their removal through dimensionality reduction via PCA can significantly reduce the condition number of the effective Hessian in the reduced space, leading to more stable and potentially faster convergence of optimization algorithms. The PCsInit method leverages this by initializing the weights in a way that aligns with the principal components, effectively operating in a space where the Hessian is better conditioned.

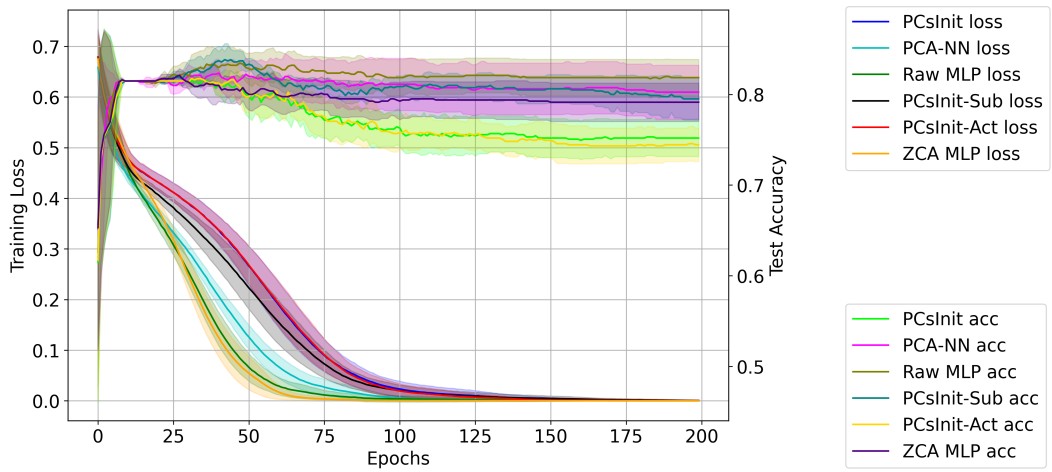

Figure 16: Heart dataset with Xavier initialization and uncertainty

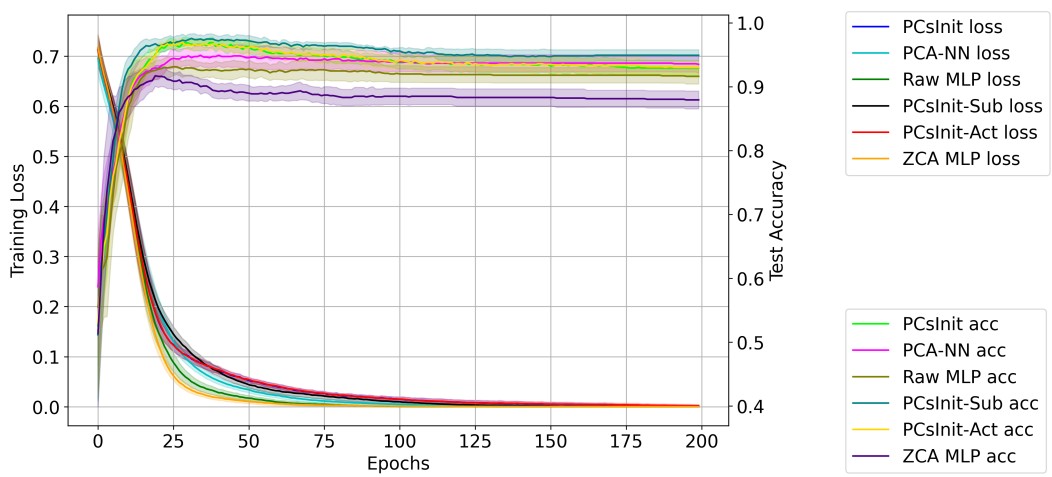

Figure 17: Ionosphere dataset with Xavier initialization and uncertainty

### A.6 PROOF FOR THEOREM 3.2

**Statement.** Let $x \in \mathbb{R}^d$ be the input vector, and let $W_r \in \mathbb{R}^{d \times r}$ be the weight matrix of the first layer, where $r \leq d$ is the number of principal components used. The columns of $W_r$ are the principal component vectors. Assuming that the bias term is zero, i.e., the output of the first layer is $h^1 = W_r^T x$. Then, the Lipschitz constant for the first layer in PCsInit is: $\sigma_{max}(W_r) = ||W_r||$.

**Proof A.2** *Let's consider the difference in the output of the first layer for two different inputs, $x$ and $y$:*

$$h^1(x) - h^1(y) = W_r^T x - W_r^T y = W_r^T (x - y).$$

*Now, take the norm of both sides:*

$$||h^1(x) - h^1(y)|| = ||W_r^T (x - y)||$$
$$\leq ||W_r^T|| \cdot ||x - y||.$$

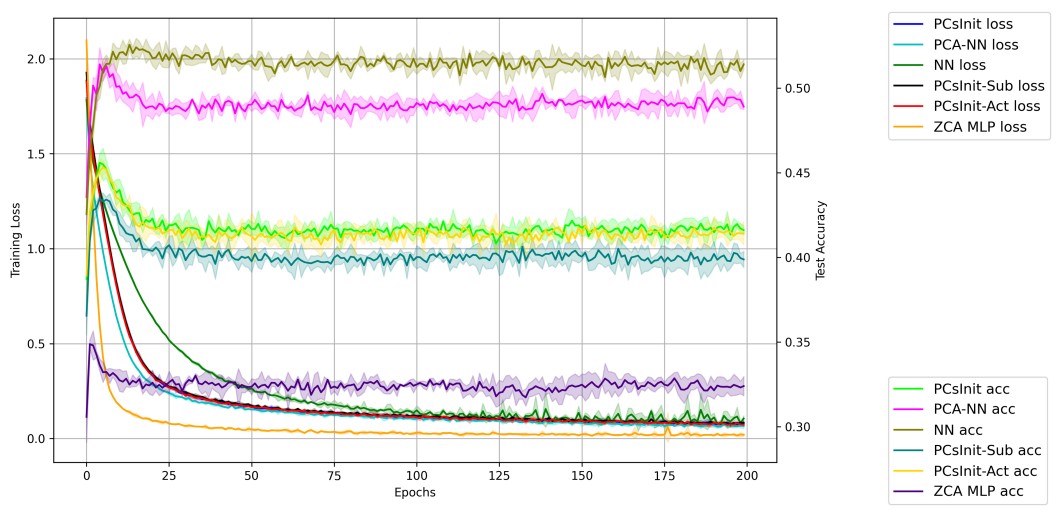

Figure 18: CIFAR-10 dataset with He initialization and uncertainty

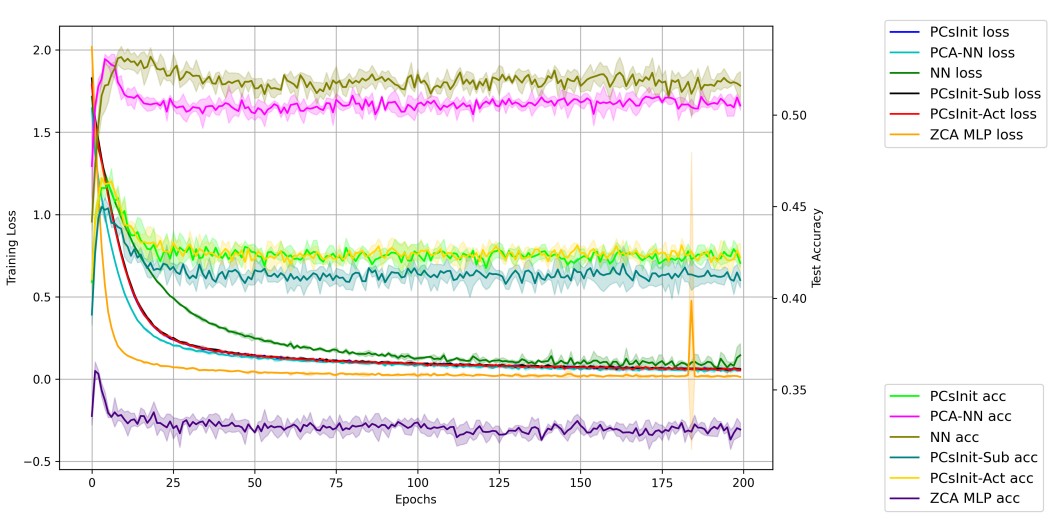

Figure 19: CIFAR-10 dataset with Xavier initialization and uncertainty

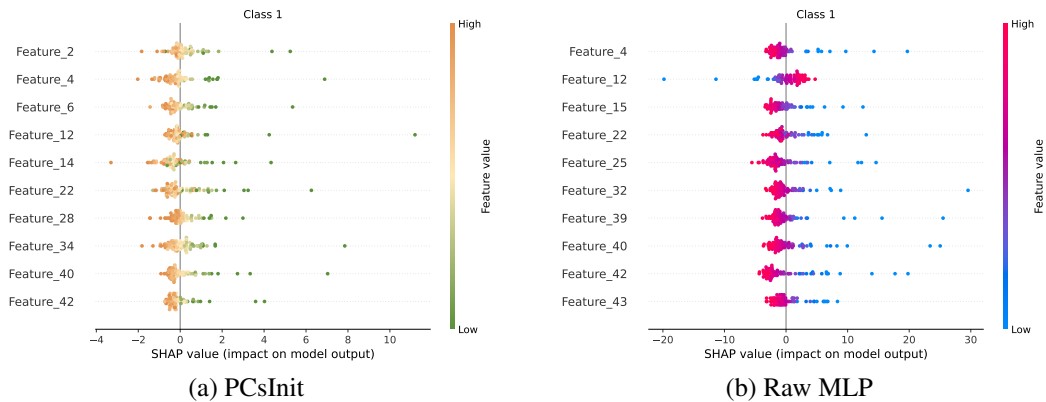

Figure 20: Top 10 most important features contributing to (a) **PCsInit** and (b) **Raw MLP** predictions on the Heart dataset for Class 1 across the test set. The first layer of PCsInit is frozen throughout the training.

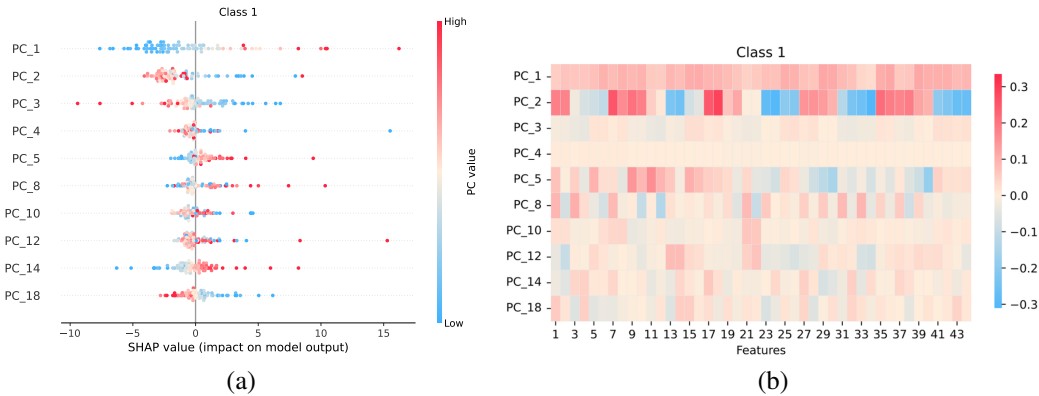

Figure 21: (a) Top 10 most principal components contributing to **PCA-NN** predictions on the Heart dataset for Class 1 across the test set. (b) Heatmap of feature importance through these components.

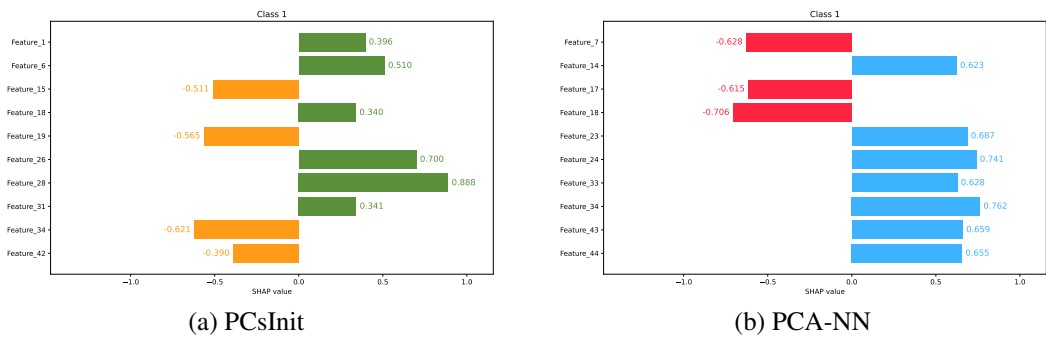

Figure 22: The local feature importance of a data point contributing to the **PCsInit** prediction and **PCA-NN** prediction through PC_2 on the Heart dataset for Class 1.

*It is known that the spectral norm of a matrix is equal to its largest singular value (Nguyen, 2022). Since the singular values of $W_r^T$ are the same as the singular values of $W_r$, we can also say that $||W_r^T|| = ||W_r||$. Let $\sigma_{max}(W_r)$ be the largest singular value of $W_r$. Then:*

$$||h^1(x) - h^1(y)|| \leq \sigma_{max}(W_r) \cdot ||x - y||.$$

Table 3: Principal angles between PCA-initialized first-layer weights and fine-tuned weights.

| Dataset | Heart | Ionosphere | Parkinson | Micromass |
|---|---|---|---|---|
| Mean Principal Angle | 6.2° | 2.9° | 31.9° | 36.8° |

*Therefore, the Lipschitz constant for the first layer in PCsInit is:*

$$L_1 = \sigma_{max}(W_r) = ||W_r||.$$

## A.7 PROOF FOR THEOREM 3.3

**Statement.** Consider the first layer of a neural network with the weight matrix $W_r \in \mathbb{R}^{d \times r}$ that consists of the first $r$ principal components (PCs) of the input data, forming orthonormal rows, i.e., $W_r W_r^T = I$. Following this linear layer, an element-wise Lipschitz continuous activation function $\sigma : \mathbb{R} \to \mathbb{R}$ with Lipschitz constant $L_\sigma$ is applied. Then, the Lipschitz constant $L_1$ of the first layer operation $f_1(x) = \sigma(W_r^T x)$ with respect to the $L_2$ norm is $L_\sigma$.

**Proof A.3** *Let $f_1 : \mathbb{R}^d \to \mathbb{R}^r$ be the operation of the first layer, defined as $f_1(x) = \sigma(W_r^T x)$, where $\sigma$ is applied element-wise. We aim to find the Lipschitz constant $L_1$ such that for all $x, y \in \mathbb{R}^d$:*

$$||f_1(x) - f_1(y)||_2 \le L_1 ||x - y||_2.$$

*Substituting the definition of $f_1(x)$:*

$$||\sigma(W_r^T x) - \sigma(W_r^T y)||_2 \le L_1 ||x - y||_2.$$

*Let $a = W_r^T x$ and $b = W_r^T y$. Then the inequality becomes:*

$$||\sigma(a) - \sigma(b)||_2 \le L_1 ||x - y||_2.$$

*Since $\sigma$ is an element-wise Lipschitz continuous activation function with Lipschitz constant $L_\sigma$, for each component $i \in \{1, ..., r\}$:*

$$|\sigma(a_i) - \sigma(b_i)| \le L_\sigma |a_i - b_i|.$$

*Squaring both sides and summing over all components:*

$$\sum_{i=1}^{r} (\sigma(a_i) - \sigma(b_i))^2 \le L_\sigma^2 \sum_{i=1}^{r} (a_i - b_i)^2.$$

*Taking the square root of both sides yields:*

$$||\sigma(a) - \sigma(b)||_2 \le L_\sigma ||a - b||_2.$$

*Substituting back $a = W_r^T x$ and $b = W_r^T y$:*

$$||\sigma(W_r^T x) - \sigma(W_r^T y)||_2 \le L_\sigma ||W_r^T x - W_r^T y||_2.$$

*By the linearity of matrix multiplication:*

$$||\sigma(W_r^T x) - \sigma(W_r^T y)||_2 \le L_\sigma ||W_r^T (x - y)||_2.$$

*Given that the weight matrix $W_r$ is formed by the first $r$ principal components of the input data, forming semi-orthonormal rows, we have $W_r W_r^T = I$. Therefore, the largest singular value (which is the spectral norm) is 1. Substituting this back into the Lipschitz inequality:*

$$||\sigma(W_r^T x) - \sigma(W_r^T y)||_2 \leq L_\sigma(1)||x - y||_2$$
$$\Rightarrow ||\sigma(W_r^T x) - \sigma(W_r^T y)||_2 \leq L_\sigma||x - y||_2.$$

*Thus, the Lipschitz constant of the first layer $f_1(x) = \sigma(W_r^T x)$ is $L_1 = L_\sigma$.*

## A.8   PROOF OF THEOREM 3.4

**Statement.** Assume that $\tilde{x} = x + \eta$, where $\tilde{x}$ is the noisy input, $x$ is the clean input, and $\eta$ is the noise vector. In addition, assume also that $\eta \sim \mathcal{N}(0, \sigma^2 I)$, i.e., the noise follows a Gaussian distribution with zero mean and covariance matrix $\sigma^2 I$. Here, $\sigma^2 \in \mathbb{R}^+$ and $I$ is the identity matrix. Next, let the eigenvalues of $X^T X$ be $\lambda_1, \lambda_2, ..., \lambda_r$. Then, for PCsInit, the noise propagated after the first layer is $W_r^T \eta$ follows a Gaussian distribution with mean 0 and its covariance matrix is a diagonal matrix with entries $\sigma^2 \lambda_1, \sigma^2 \lambda_2, ..., \sigma^2 \lambda_r$.

**Proof A.4** *The output of the first layer is*

$$h^1 = W_r^T \tilde{x} \tag{3}$$
$$= W_r^T(x + \eta) = W_r^T x + W_r^T \eta. \tag{4}$$

*Since $\eta \sim \mathcal{N}(0, \sigma^2 I)$, the transformed noise $W_r^T \eta$ will also be Gaussian with mean:*

$$\mathbb{E}[W_r^T \eta] = W_r^T \mathbb{E}[\eta] = W_r^T 0 = 0. \tag{5}$$

*and covariance matrix:*

$$Cov(W_r^T \eta) = \mathbb{E}[(W_r^T \eta - 0)(W_r^T \eta - 0)^T] = W_r^T \mathbb{E}[\eta \eta^T] W_r \tag{6}$$
$$= W_r^T(\sigma^2 I)W_r = \sigma^2 W_r^T W_r. \tag{7}$$

*Therefore, $W_r^T \eta \sim \mathcal{N}(0, \sigma^2 W_r^T W_r)$.*

*Since $W_r$ comes from the selected $r$ eigenvalues - eigenvectors pair of PCA, $W_r^T W_r$ is a diagonal matrix with the eigenvalues of $X^T X$ on the diagonal. Therefore, the covariance matrix of $W_r^T \eta$ is a diagonal matrix with entries $\sigma^2 \lambda_1, \sigma^2 \lambda_2, ..., \sigma^2 \lambda_r$.*

## A.9   PROOF OF THEOREM 3.5

**Statement.** Consider a neural network where the first layer's weight matrix, $W_r \in \mathbb{R}^{d \times r}$, is initialized as in PCsInit, and therefore $W_r$ is orthonormal. Suppose that the input is corrupted by additive white noise, i.e., $\tilde{x} = x + \eta$, where $x \in \mathbb{R}^d$ is the clean input signal and $\eta \sim \mathcal{N}(0, \sigma^2 I)$ is the additive white noise. Then, the norm of the noise component after the first layer is preserved, i.e., $||W_r^T \eta|| = ||\eta||$. Also, let $f : \mathbb{R}^d \to \mathbb{R}^r$ be the neural network function, decomposed into layers $f = f_L \circ f_{L-1} \circ ... \circ f_1$, where $f_i$ represents the $i$-th layer, $f(x)$ is the clean output, and $\tilde{f}(\tilde{x})$ is the noisy output. Further assume that each subsequent layer $f_i$ for $i = 2, ..., L$ is $L_i$-Lipschitz continuous. Then, $||\tilde{f}(\tilde{x}) - f(x)|| \leq \left(\prod_{i=2}^L L_i\right)||\eta||$.

**Proof A.5** *For PCsInit, the output of the first layer with noisy input is:*

$$\tilde{h}_1 = f_1(\tilde{x}) = W_r^T(x + \eta) = W_r^T x + W_r^T \eta = h_1 + W_r^T \eta$$

*Since $W_r = Q$ (where $Q \in \mathbb{R}^{d \times r}$ is semi-orthonormal, meaning $Q^T Q = I_r$, so $W_r^T W_r = I_r$), and thus $W_r^T$ (an $r \times d$ matrix) has semi-orthonormal rows ($W_r^T(W_r^T)^T = W_r^T W_r = I_r$), so $||W_r^T|| = 1$. This implies that the noise component's norm is preserved:*

$$||W_r^T \eta|| = ||\eta||$$

*Using the Lipschitz property of subsequent layers, we can bound the noise propagation:*

$$||\tilde{f}(\tilde{x}) - f(x)|| = ||f_L(f_{L-1}(...f_1(\tilde{x}))) - f_L(f_{L-1}(...f_1(x)))||$$
$$\leq L_L L_{L-1}...L_2 ||f_1(\tilde{x}) - f_1(x)||$$
$$\leq \left( \prod_{i=2}^{L} L_i \right) ||\eta||$$

## A.10  PROOF OF THEOREM 3.6

**Statement.** For any layer $\ell > 1$, let $\rho^\ell$ is the activation function, $W^\ell \in \mathbb{R}^{d_l \times d_{l-1}}$ is the weight matrix, and $b^\ell \in \mathbb{R}^{d_l}$ is the bias vector. Then, the output at layer $\ell$ is: $h^\ell = \rho^\ell(W^\ell h^{\ell-1} + b^\ell)$. Next, let $\eta^\ell$ represents the noise in the output of layer $\ell$ and assume also that $\rho^\ell$ is Lipschitz continuous with Lipschitz constant $L_\ell$, i.e. $||h^\ell - \tilde{h}^\ell|| \leq L_\ell ||W^\ell|| \cdot ||\eta^{\ell-1}||$, where $\tilde{h}^\ell$ is the clean output and $h^\ell$ is the noisy output. Then, the bound for the noise propagated to the second layer is $||\eta^2|| \leq L_2 ||W^2|| \cdot ||W^1|| \cdot ||\eta^0||$. In addition, the general noise bound for any layer $\ell > 1$ is: $||\eta^\ell|| \leq \left[ \prod_{i=2}^{l} (L_i ||W^i||) \right] \cdot ||W^1|| \cdot ||\eta^0||$.

**Proof A.6** *Substituting the noisy input from the previous layer gives*

$$h^\ell = \rho^\ell(W^\ell(h^{\ell-1} + \eta^{\ell-1}) + b^\ell) = \rho^\ell(W^\ell h^{\ell-1} + b^\ell + W^\ell \eta^{\ell-1}) \tag{8}$$

*We first examine the **noise propagation to layer 2**:*

*Let $\eta^0 = \eta$ be the noise added to the input: $\tilde{x} = x + \eta$. The output of the first layer with noisy input is:*

$$h^1 = W^1 \tilde{x} = W^1(x + \eta) = W^1 x + W^1 \eta$$

*The output of the first layer with clean input is:*

$$\tilde{h}^1 = W^1 x$$

.

*Therefore, the noise at the output of layer 1 is:*

$$\eta^1 = h^1 - \tilde{h}^1 = W^1 \eta \tag{9}$$

*The output of the second layer with noisy input from the first layer is:*

$$h^2 = \rho^2(W^2 h^1 + b^2) = \rho^2(W^2(W^1 x + W^1 \eta) + b^2).$$

*The output of the second layer with clean input from the first layer is:*

$$\tilde{h}^2 = \rho^2(W^2 \tilde{h}^1 + b^2) = \rho^2(W^2(W^1 x) + b^2).$$

*Therefore, the noise at the output of layer 2 is:*

$$\eta^2 = h^2 - \tilde{h}^2 = \rho^2(W^2 h^1 + b^2) - \rho^2(W^2 \tilde{h}^1 + b^2) = \rho^2(W^2(W^1 x + W^1 \eta) + b^2) - \rho^2(W^2(W^1 x) + b^2)$$

*Using the Lipschitz property of the activation function $\rho^2$ (with Lipschitz constant $L_2$), we have:*

$$||\eta^2|| = ||h^2 - \tilde{h}^2|| \leq L_2 ||W^2(h^1 - \tilde{h}^1)|| = L_2 ||W^2 \eta^1|| \leq L_2 ||W^2|| \cdot ||\eta^1||$$

*Substituting the noise at the output of layer 1 in 9:*

$$||\eta^2|| \le L_2||W^2|| \cdot ||W^1\eta|| \le L_2||W^2|| \cdot ||W^1|| \cdot ||\eta||$$

*The difference (noise) in the output of layer $\ell$ is $\eta^\ell = h^\ell - \tilde{h}^\ell$ and we can apply recursive application to derive its bound. Recall that for layer 1, we already established that:*

$$||\eta^1|| = ||h^1 - \tilde{h}^1|| \le ||W^1|| \cdot ||\eta^0||$$

*Next, for layer 2:*

$$||\eta^2|| \le L_2||W^2|| \cdot ||W^1|| \cdot ||\eta^0||$$

*Now, for layer 3:*

$$||\eta^3|| = ||h^3 - \tilde{h}^3|| \le L_3||W^3|| \cdot ||\eta^2||.$$

*Hence, substituting the bound for $||\eta^2||$ we have:*

$$||\eta^3|| \le L_3||W^3|| \cdot (L_2||W^2|| \cdot ||W^1|| \cdot ||\eta^0||) = L_3||W^3|| \cdot L_2||W^2|| \cdot ||W^1|| \cdot ||\eta^0||$$

*By continuing this pattern, we get for any layer $\ell > 1$:*

$$||\eta^\ell|| \le (L_\ell||W^\ell||) \cdot (L_{\ell-1}||W^{\ell-1}||) \cdot ... \cdot (L_2||W^2||) \cdot ||W^1|| \cdot ||\eta^0||$$

*For the output layer (layer L), the bound becomes:*

$$||\eta^L|| \le (L_L||W^L||) \cdot (L_{L-1}||W^{L-1}||) \cdot ... \cdot (L_2||W^2||) \cdot ||W^1|| \cdot ||\eta^0||$$

