# OpenReview forum: "Principal Components for Neural Network Initialization: A Novel Approach to Explainability and Efficiency"
_ICLR.cc/2026/Conference — ICLR 2026 Conference Withdrawn Submission_

### Official Review · Reviewer_SGM9 · 2025-10-28

**Soundness:** 3
**Presentation:** 2
**Contribution:** 3
**Rating:** 4
**Confidence:** 3

**Summary:**

This paper proposes Principal Components-based Initialization (PCsInit) and two extensions (PCsInit-Act and PCsInit-Sub) as new weight initialization methods for neural networks, inspired by the structure of Principal Component Analysis (PCA). The central claim is that initializing the first layer with PCA-derived principal components leads to better optimization stability, improved explainability for XAI methods (e.g., SHAP, LIME), and potentially better generalization compared to models trained directly on PCA-reduced data (PCA-NN). The authors provide theoretical derivations regarding Hessian conditioning, Lipschitz continuity, and robustness to noise, as well as experiments on several datasets (Heart, Parkinson, Micromass, HTAD, MNIST, etc.) to support their claims.

**Strengths:**

[1] The paper explores an original direction in connecting PCA with neural network initialization rather than data preprocessing, which could have practical implications for explainable AI.
[2] The presentation of theoretical properties (e.g., conditioning, Lipschitz constant) provides mathematical intuition for the potential advantages of PCA-based initialization.
[3] The methods are computationally lightweight, potentially easy to reproduce.

**Weaknesses:**

[1] The paper claims that PCsInit “improves interpretability” compared to PCA-NN, yet all empirical evidence focuses on SHAP value visualization differences. There is no quantitative or formal evaluation of interpretability, nor a clear metric to support that the method meaningfully improves explanation fidelity.
[2] The figures show SHAP values for PCA-NN and PCsInit but do not explain why one visualization is “better.” The visual difference in SHAP plots does not constitute a formal improvement in interpretability. Moreover, SHAP itself is an approximation, and no quantitative assessment is provided.
[3] The neural network architectures are only vaguely described (“five layers”) without specifying activation types, layer widths, or normalization techniques.
[4] The datasets used are relatively small and scalability to large models or complex data remains untested.
[5] The paper mentions in Section 4.2 that the first r components are retained, but does not specify how r is chosen or whether variance thresholds are used. Since the number of components directly affects initialization rank and model expressivity, an ablation varying r would clarify sensitivity and optimality.
[6] The paper omits architectural and training details crucial for reproducibility, such as activation functions, learning rate, batch size, and optimizer type.

**Questions:**

[1] Does freezing and unfreezing the first layer meaningfully affect convergence compared to continuous fine-tuning?
[2] Most datasets are small tabular ones (Heart, Parkinson, Micromass). What motivated their choice, and have any larger or more complex datasets been tested to confirm scalability?
[3] The SHAP and LIME plots are visually different, but the evaluation is qualitative. Could the authors provide numerical interpretability metrics to support the claim that PCsInit improves explainability?
[4] The motivation for these two variants is described conceptually but not evaluated in depth. What specific empirical improvements do they bring relative to base PCsInit?

---

> ### Author Response · Authors · 2025-11-21
> **Response to Reviewer SGM9 - part I**
>
> We thank you for your valuable feedback. We have completed a major revision to address all your concerns:
>
> 1.  **On Quantitative Interpretability (Weakness 1, 2; Question 3):**
>
> Our revised paper now includes a **quantitative XAI metrics**. We measure **faithfulness** (using AOPC), **stability** (using variance of the explanation across 10 seeds), and **fidelity** (using synthetic datasets) for both PCsInit and Raw MLP. These results formally demonstrate that PCsInit provides more reliable explanations.
>
>   - **Faithfulness:** Faithfulness is evaluated via the **Area Over the Perturbation Curve (AOPC)**, computed by zero-perturbation of features in Most Relevant First (MoRF) order. Figure 4 presents the AOPC distributions obtained by removing the top 1 to 5 Most Relevant First (MoRF) features, ordered by their absolute SHAP values, using zero perturbation across 10 runs on the Heart dataset. Overall, while both PCsInit and Raw MLP yield positive AOPC scores for class 0, their behaviors diverge for class 1: PCsInit remains positive, whereas Raw MLP becomes negative. This indicates that the top-ranked features consistently play a meaningful role in the PCsInit’s predictions for both classes.
>
>   - **Stability:** It is quantified using the variance of the explanation values across runs, with lower variance indicating greater stability. Across 10 runs with different random seeds, SHAP achieves mean variance scores of 0.622 and 1.130 under PCsInit and Raw MLP, respectively, highlighting the higher stability of PCsInit explanations.
>
>   - **Fidelity:** To evaluate fidelity, we generate five synthetic binary classification datasets following the *ssin* synthetic design used for classification in reference (4), each containing 2000 samples and 20 features. The number of predefined important features that directly influence the target ranges from 3 to 7 across the five datasets, while the remaining features are irrelevant or noisy (i.e., the first dataset has 3 important features, the second has 4, and so on). We then compare the top important features identified by SHAP under the PCsInit and Raw MLP models with the ground-truth important features. The mean recovery scores of both methods are above 80\%, demonstrating high fidelity in capturing the underlying ground-truth feature structure.
>
> References:
>
> (1) Evaluating the visualization of what a Deep Neural Network has learned https://arxiv.org/abs/1509.06321
>
> (2) Explainable artificial intelligence: A survey of needs, techniques, applications, and future direction https://arxiv.org/abs/2409.00265
>
> (3) Assessing Fidelity in XAI post-hoc techniques: A Comparative Study with Ground Truth Explanations Datasets https://arxiv.org/pdf/2311.01961
>
> (4) Using Sensitivity Analysis and Visualization Techniques to Open Black Box Data Mining Models https://scispace.com/pdf/using-sensitivity-analysis-and-visualization-techniques-to-2xqx36gbu2.pdf

---

> ### Author Response · Authors · 2025-11-21
> **Response to Reviewer SGM9 - part II**
>
> 2.  **On Reproducibility (Weakness 3, 6):** We apologize for omitting crucial details.
>
>
>   We employed a consistent experimental configuration across all datasets to ensure fair comparison. The baseline and proposed models utilized a 5-layer fully connected architecture. Optimization was performed using Adam  with a static learning rate of $0.01$. The batch size was fixed at 64 samples. For the standard PCsInit method, the first layer utilizes a linear identity pass (no activation) to strictly mimic linear PCA projection at initialization, whereas subsequent layers utilize ReLU activations. We have updated these details to Appendix A.1. We have provided an **anonymized link to our code in the paper**. You can also access it at https://anonymous.4open.science/r/pcsinit-0675/README.md
>
> **On Freezing/Unfreezing the first layer (Question 1):**
>
> *Freezing the first layer for an initial period (for example, 30 epochs) prevents drastic weight updates in the first layer immediately after initialization, ensuring the principal components provide a meaningful starting point, and allows the deeper (subsequent) layers to adapt to the PCA-based feature space and establish stable higher-level representations before the first layer is allowed to drift.* We have updated the manuscript to clarify this point.
>
>
>
> **On Datasets and Variants (Weakness 4, Questions 2, 4):**
>
> -We have added the CIFAR-10 datasets. This added the variety to the characteristics of datasets used: small, large, noisy and imbalanced. We also have strengthened the analysis by adding several **stronger baselines** (standard MLP on raw data,  ZCA whitening).
>
> - We have also added more analysis to clarify the empirical improvements of the variants (e.g., PCsInit-Act's strong performance on noisy data, and PCsInit-Sub's computational trade-off).
>
> **On the choice of $r$ (Weakness 5)**
> The dimensionality of the principal component space, **$r$, was determined dynamically for each dataset based on the explained variance ratio**. We selected the minimum number of components required to explain $95\%$ of the cumulative variance in the training data, which is commonly used in some other studied as cited in Appendix A.1. Consequently, the width of the neural network's hidden layers was set equal to $r$ to maintain consistency with the intrinsic dimensionality of the input.
>
> **The choice of $r$** affects the percentage of variance explained (which implies the amount of information retained) after PCA. *Therefore, increasing $r$ will likely increase the performance of the model to a certain threshold (for example, 95\% or 99\% of variance explained). However, a very high $r$ may lead to retaining noise, which may decrease the performance of the model, especially for noisy data.*
>
> We hope these revisions, especially the new quantitative XAI and the reproducibility appendix, have fully addressed your concerns. Also, we believe these substantial revisions have significantly strengthened the paper's rigor and validity. We kindly ask that you reconsider your score in light of these improvements.

---

> ### Comment · Reviewer_SGM9 · 2025-11-25
>
> Well received the improvements. I will increase the rate to 6.

---

> > ### Author Response · Authors · 2025-11-25
> > **Response to Reviewer SGM9**
> >
> > Thank you very much. We are glad that we have addressed your concerns. And thank you again for your constructive comments, which has significantly strengthened the paper.

---

### Official Review · Reviewer_5xdt · 2025-10-31

**Soundness:** 3
**Presentation:** 3
**Contribution:** 3
**Rating:** 8
**Confidence:** 3

**Summary:**

The authors propose a principal components-based initialization strategy to incorporate PCA into the first layer of a neural network via initialization of the first layer in the network with the principal components.

**Strengths:**

- The paper is well organized and written.
- The contribution is theoretically interesting
- The proposed method named  PCsInit is detailed and reproducible.
- Experiments show the performance of PCsInit on the Heart dataset.

**Weaknesses:**

Comparison should be extended with more previous algorithms.

**Questions:**

Is it possible to design a sparse version of the PCsInit to reduce computational time?

---

> ### Author Response · Authors · 2025-11-21
> **Response to Reviewer 5xdt**
>
> We sincerely thank you for your positive and encouraging review. We are glad you found the paper well-organized and the contribution "theoretically interesting."
>
> We have acted on your primary piece of feedback regarding the experimental comparisons.
>
> 1.  **On Extending Comparisons (Weakness):** We have taken this feedback and **significantly expanded our experimental section**. Our revised paper now includes several stronger baselines across all datasets:
>     * A standard MLP trained on raw, standardized data.
>     * A ZCA whitening pre-processing baseline.
>     This provides a much more comprehensive evaluation of PCsInit's performance.
>
> 2.  **On a Sparse Version (Question):** This is a very interesting idea. A sparse version of PCsInit could indeed be designed, for example, by using **Sparse PCA** (which often uses an L1 penalty) instead of standard PCA to generate the initial weights for the first layer. This would not only reduce computational time (by creating sparse weight matrices) but could also potentially improve interpretability, as each component would be defined by a smaller, less dense set of original features. We believe this is a promising direction for future research and have **added it to our conclusion** as such.
>
> In addition, we also revised the paper with more evaluation metrics, and addressed the comments from the other reviewers. We hope these revisions reinforce your positive assessment and justify your strong rating.
> Thank you again for your support of our work.

---

### Official Review · Reviewer_q5ia · 2025-11-01

**Soundness:** 2
**Presentation:** 2
**Contribution:** 2
**Rating:** 2
**Confidence:** 4

**Summary:**

They propose a new machine learning framework called Principal Components–based Initialization (PCsInit), which initializes a network’s first layer with PCA loadings and adds two variants (PCsInit-Act and PCsInit-Sub) to preserve interpretability while enabling nonlinear modeling.
The approach aims to make feature attributions more transparent than PCA-preprocessing pipelines and, across real-world classification/regression benchmarks, matches or outperforms PCA-NN baselines with simpler explanations.

**Strengths:**

**(S1).** It removes explainability constraints common to PCA-based baselines by tying the first-layer weights directly to principal components, preserving a clear input - PC - prediction mapping.

**(S2).** It introduces a practical variants—PCsInit-Act—to better capture nonlinear patterns adding activations function after PC.

**Weaknesses:**

**(W1)**. The first-layer weights initialized with PCA are free to drift during training, so $W^{\text{optimal}}_1$ cannot be said to span the same subspace as the top-$r$ eigenvectors; to substantiate the interpretability claim, please either (i) prove or measure subspace closeness ( e.g., principal angles or $|W^{\top}_1 W_r|_F$ ) across epochs, or (ii) show that projection by $W^{\text{optimal}}_1$ preserves variance essentially as well as $W_r$ on data.

**(W2).** Section 3’s bounds appear to concern estimation/generalization error rather than optimization dynamics; they do not formally imply faster or more stable convergence. If efficiency is a key claim, provide optimization guarantees (e.g., convergence rate lower bounds or excess-risk upper bounds as a function of $(n, r)$ ) that specifically distinguish PCsInit from PCA-preprocessing baselines.

**(W3).** To justify the method as “explainable,” feature importance produced by PCsInit should be directionally consistent with the feature importance from other XAI methods or that it reliably identifies truly important variables. To this end, go beyond PCA-NN and experimentally compare feature-importance rankings against multiple XAI baselines, or verify on synthetic datasets with known ground truth that significant variables are accurately recovered.

**(W4).** The computational-efficiency story is under-evidenced. Please add training time vs epoch curves for PCsInit, PCA-NN, and PCsInit-Sub, include PCA/SVD setup cost; current experiments do not isolate whether PCsInit-sub actually reduces total training time for comparable accuracy.

**(W5).** Because interpretability hinges on the PCA semantics of the first layer, quantify attribution drift: compare per-PC attributions at initialization vs. at convergence, and report whether explanation faithfulness degrades as $W_1$ departs from the PCA subspace (e.g., sparsity/orthogonality metrics, input reconstruction error through $W_1$).

**Questions:**

**(Q1).** Could you standardize the rule for choosing the first-layer width? In this paper, $r$ is usually used as a dimension of reduced space. The text alternates between “retain n components” and “retain r% of variance,” while the appendix mentions “at least 95%”.

**(Q2).** In Algorithm 3 you list “ReLU, He, …” as the activation after the PCA-initialized layer, but “He” is an initialization, not an activation. Which activations can be actually used in PCsInit-Act?

**(Q3).** How are $J(\cdot)$ and the Hessian matrix defined for $r>1$? In Theorem 3.1, you set $J(W)=\tfrac{1}{2}\lVert W^{\top}X - Y\rVert^{2}$, but when $r>1$ and $W_r\in\mathbb{R}^{d\times r}$ is substituted, a dimensional mismatch occurs. Therefore, it seems necessary to review the statement of Theorem 3.1 overall.

**[Minor]**
1. Expression (LaTeX) :  In p.3, “so that **n** components is retained” → “so that **$n$** components is retained” ?

2. Expression (grammar): In p.7, “the results of performance loss and accuracy **is** reported” → “the results of performance loss and accuracy **are** reported ” ?

3. Expression (grammar): In p.7,  “**performances** of the proposed techniques” → “**performance** of the proposed techniques” ?

---

> ### Author Response · Authors · 2025-11-21
> **Response to Reviewer q5ia**
>
> We thank you for your time and your insightful feedback, especially regarding the theoretical validity and the "weight drift" of our method. These points were critical and have led to significant improvements in our paper. We have completed a revision to address your concerns:
>
> **Questions Q1**
> We are sorry about the typos on n being the number of components. To standardize the initialization criteria across diverse datasets, the width of the first layer is rigorously defined by the data's spectral properties. Specifically, **the layer width corresponds to the number of principal components $r$ satisfying a cumulative explained variance threshold of $0.95$**. This ensures that the network capacity scales adaptively with the information content of the dataset.
> **The choice of $r$** affects the percentage of variance explained (which implies the amount of information retained) after PCA. *Therefore, increasing $r$ will likely increase the performance of the model to a certain threshold (for example, 95\% or 99\% of variance explained). However, a very high $r$ may lead to retaining noise, which may decrease the performance of the model, especially for noisy data.*
>
> Regarding the **bias and variance trade off**, for PCsInit and its variants, the number of retained principal components, **$r$, serves as a crucial structural hyperparameter that directly governs the model's bias-variance tradeoff**. When $r$ is set to a small value, the first layer acts as a narrow information bottleneck, aggressively filtering out input dimensions with lower eigenvalues. This regime typically results in high bias and low variance: the model is stable and robust to noise because it ignores the idiosyncratic fluctuations of the data, but it risks underfitting if the target variable relies on subtle, fine-grained features that were discarded during the dimension reduction.
>
> Conversely, selecting a large value for $r$ (approaching the total number of features) preserves nearly all the information present in the input, shifting the model towards a low bias and high variance state. In this regime, the initialization captures complex relationships and fine details, minimizing systematic error. However, because the lower-variance components often contain a lower Signal-to-Noise Ratio (SNR), a large $r$ allows the network to ingest and learn from random noise patterns, making the model more susceptible to overfitting and less consistent when generalizing to unseen data.
>
>
>
> **Questions Q2**
> The typo in Algorithm 3 ("He" as an activation) has been corrected. For the PCsInit-Act variant, we introduced non-linearity immediately following the PCA-initialized layer.  **The method is not restricted to a specific activation and can work with common activation functions (including ReLU and LeakyReLU,...)**, the results reported in this study utilize the Rectified Linear Unit (ReLU) activation function.
>
> **On Theoretical Rigor (Weakness W2, Question Q3):**
> We are sorry for the confusing notation and typos. The notation $W$ is the weights and are not related to $W_r$. We have **updated the proof and statement of Theorem 3.1** to with an alternative notation ($V$) and corrected some typos in the theorem.
>
> Also, we have added a citation from *Classical and modern numerical analysis: Theory, methods and practice* by Ackleh, Azmy S and Allen, Edward James and Kearfott, R Baker and Seshaiyer, Padmanabhan, which support that **for nonlinear optimization is highly sensitive to scaling and
> the problem’s condition number, and that convergence can be
> significantly slow when the condition number of the Hessian
> matrix is large.** We hope this clarified that out statements provides *intuition* for why starting with a better-conditioned space is beneficial, rather than serving as a formal proof of faster convergence for the full, non-linear network.
>
> **On Computational Efficiency (Weakness W4):**
>
> The computational cost analysis presented in Figure 18 has been updated to include the one-time setup cost associated with the eigenvalue decomposition (PCA fitting). While PCsInit incurs a higher per-epoch cost during the fine-tuning phase compared to PCA-NN (which operates on reduced input dimensions), the PCsInit-Sub variant significantly mitigates the initial setup overhead by estimating principal components on a random data subset, offering a balanced trade-off between initialization speed and asymptotic convergence.

---

> ### Author Response · Authors · 2025-11-21
> **Response to Reviewer q5ia - part II**
>
> **On Weight Drift (Weakness W1, W5):**
> We realized that the writing of the paper has not been clear, and therefore we we have highlighted this in the abstract, contribution, experiments of the revised manuscript that: **for the PCsInit approach, if the weights of the first layer are frozen throughout the training process then its performance is the same as applying PCA to the input data and then training a model on the principal components. However, when the first layer is fine-tuned, they are not equivalent.**  However, PCsInit can still be considered as a dimension reduction method (we added a paragraph on **Dynamics of Fine-Tuning and weight drifts from standard PCA** in the Methodologies section to explain more detailed).
>
> Moreover, we have added **"attribution drift"** by calculating the **subspace similarity** (using principal angles) between the initial $W_r$ matrix and the final, fine-tuned $W_1^{optimal}$ matrix. This allows us to quantify how much the weights depart from the original PCA subspace, when PCsInit has a fine-tuning phase. Principal angles for the Heart and Ionosphere datasets are in turn $6.2^\circ$ and $2.9^\circ$, demonstrating the first-layer weights hardly change subspace during training. This means that the PCA initialization captures the main structure of the data. However, larger angles for Parkinson ($31.9^\circ$) and Micromass ($36.8^\circ$) imply that the model can change its direction to optimize performance.
>
> **On XAI Quantification (Weakness W3):** We agree that our explainability claim was under-evidenced. We have added a paragraph with **quantitative XAI metrics** for PCsInit and NN explanations.
> - To formally validate our method, we have **added a new experiment on a synthetic dataset with known ground-truth feature importance**. We generate five synthetic binary classification datasets following the *ssin* synthetic design used for classification in reference (4), each containing 2000 samples and 20 features. The number of predefined important features that directly influence the target ranges from 3 to 7 across the five datasets, while the remaining features are irrelevant or noisy (i.e., the first dataset has 3 important features, the second has 4, and so on). We then compare the top important features identified by SHAP under the PCsInit and Raw MLP models with the ground-truth important features. The mean recovery scores of both methods are above 80\%, demonstrating high fidelity in capturing the underlying ground-truth feature structure.
>
> - We have also added standard faithfulness (AOPC) and stability metrics to further support this claim.
>
>     - **Faithfulness:** Faithfulness is evaluated via the **Area Over the Perturbation Curve (AOPC)**, computed by zero-perturbation of features in Most Relevant First (MoRF) order. Figure 4 presents the AOPC distributions obtained by removing the top 1 to 5 Most Relevant First (MoRF) features, ordered by their absolute SHAP values, using zero perturbation across 10 runs on the Heart dataset. Overall, while both PCsInit and Raw MLP yield positive AOPC scores for class 0, their behaviors diverge for class 1: PCsInit remains positive, whereas Raw MLP becomes negative. This indicates that the top-ranked features consistently play a meaningful role in the PCsInit’s predictions for both classes.
>
>     - **Stability:** It is quantified using the variance of the explanation values across runs, with lower variance indicating greater stability. Across 10 runs with different random seeds, SHAP achieves mean variance scores of 0.622 and 1.130 under PCsInit and Raw MLP, respectively, highlighting the higher stability of PCsInit explanations.
>
> References:
>
> (1) Evaluating the visualization of what a Deep Neural Network has learned https://arxiv.org/abs/1509.06321
>
> (2) Explainable artificial intelligence: A survey of needs, techniques, applications, and future direction https://arxiv.org/abs/2409.00265
>
> (3) Assessing Fidelity in XAI post-hoc techniques: A Comparative Study with Ground Truth Explanations Datasets https://arxiv.org/pdf/2311.01961
>
> (4) Using Sensitivity Analysis and Visualization Techniques to Open Black Box Data Mining Models https://scispace.com/pdf/using-sensitivity-analysis-and-visualization-techniques-to-2xqx36gbu2.pdf
>
>
>
>
>
>
> **On Minor Corrections:** We have fixed these errors.
> - The rule for choosing `r` is now standardized (95% variance) and stated consistently.
> - The grammatical errors ("is/are") have been corrected.
> - In addition, we have done throughout checking and fixed the remaining grammatical issues and typos.
>
> We believe these substantial revisions, guided by your feedback, have significantly strengthened the paper's contributions and rigor. We kindly ask that you reconsider your score in light of these improvements.

---

> > ### Author Response · Authors · 2025-11-26
> > **Response to Reviewer q5ia**
> >
> > As the discussion period is drawing to a close, we wanted to briefly check if our previous responses and the revised manuscript fully addressed your concerns. We are happy to provide any further clarifications if needed.
> > Thank you very much.

---

### Official Review · Reviewer_GuXF · 2025-11-02

**Soundness:** 1
**Presentation:** 2
**Contribution:** 2
**Rating:** 4
**Confidence:** 4

**Summary:**

The paper proposes PCsInit, which initializes the first linear layer with the top-r PCA directions, along with two variants. PCsInit-Act adds a nonlinearity after that layer; PCsInit-Sub computes PCs on a data subset to cut cost. The idea aims to keep PCA “inside” the network so explanations (e.g., SHAP/gradients) can be computed directly in input space without back-projecting from PCA features. Claimed theory covers Hessian conditioning, Lipschitz constants for the first layer, and noise propagation; experiments compare against “PCA-NN” across several tabular datasets and MNIST (with noise).

**Strengths:**

The paper’s main strength is a simple, practical idea with clear motivation.  Initialize the first layer with data PCs so the network “absorbs” a common preprocessing step while keeping explanations in the input space. This yields conceptual originality relative to the usual PCA before NN pipeline by unifying feature extraction and learning, and the two variants (subset PCA for scalability; post-activation for expressivity) make the recipe broadly usable. The approach is easy to implement, cheap to run, and aligns with a sensible inductive bias (early directions capture most of the variance, improving stability and noise handling) that is corroborated by consistent trends across several tabular datasets and a noisy-image toy case. The method is straightforward to reproduce, and the interpretability of SHAP/gradient attributions does not require back-projection. If validated more broadly, this could become a lightweight practice for tabular models (and potentially inspire analogous “in-network PCA”  in other modalities), offering a small but functional gain in robustness and a cleaner interpretability pipeline

**Weaknesses:**

The paper’s main weaknesses are (i) theoretical inconsistencies, the noise propagation and norm-preservation rely on ambiguous matrix shapes/orthogonality, and in places are improper or rely on restating standard spectral-norm/Lipschitz facts. (ii) limited and unconvincing experiments. Small tabular sets plus a toy vision case, key results relegated to the appendix, and no confidence intervals, multi-seed variance, or significance tests. (iii) weak baseline coverage—comparisons omit strong standards such as raw-input MLP/CNN with modern init+normalization, whitening/ZCA, random orthogonal/SVD inits, and representation-learning baselines.  (iv) missing ablations/sensitivity and no systematic study of the number of PCs, freeze duration, learning-rate/regularization, subset-PCA sampling, or the PCsInit-Act variant. (v) unclear scope for spatial data and flatten-then-PCA discards structure, so claims for vision models are unsupported; (vi) unmeasured interpretability, no quantitative evaluation of attribution faithfulness/stability. And (vii) reproducibility/presentation gaps, no code, typos/notation. The authors should expand the benchmarks with full tables and CIs across many seeds, add strong baselines, and include targeted ablations. Either limit claims to tabular data or provide a conv-friendly variant with CIFAR/ImageNet-style results. Evaluate attribution faithfulness/stability, and release code and standardize notation.

**Questions:**

1. The noise-handling and “norm preservation” claims seem to rely on mixed assumptions. Please state the exact assumptions (how inputs are standardized, what “orthonormal” means here, and the shape of the matrix) and provide a small synthetic check (e.g., Gaussian noise through your first layer) that reproduces your claims.

	2. Do you center/standardize features before computing PCs and during training? If not, why? Please clarify the full pipeline so others can reproduce identical PCs and results.

	3. Number of PCs (r): How is r chosen (fixed, variance-explained threshold, tuned on validation)? Show sensitivity curves for accuracy vs. r and discuss the bias–variance trade-off.

	4. How long is the first layer kept frozen, and how sensitive are results to this choice? Please add an ablation sweeping freeze duration and show convergence/learning-curve plots.

	5. Add standard baselines, (i) raw-input MLP/CNN with modern initialization + BN/LN, (ii) whitening/ZCA pipelines, (iii) random orthogonal/SVD initializations, (iv) a simple representation-learning baseline (e.g., autoencoder).

	6. Run more seeds and report means ± confidence intervals and significance tests. For tabular, consider a standard suite. Include CIFAR-10/100 with convnets (or explain why the method is tabular-only).

	7. The paper argues for cleaner attributions because explanations stay in the input space. Please quantify this using standard tests (faithfulness via deletion/insertion/AOPC, stability across seeds) and show whether your method alters conclusions from SHAP/IG relative to PCA-NN and raw-input models.

	8. How are subsets sampled, and what is the accuracy vs. compute trade-off? Compare to streaming/incremental PCA and show when subset PCA breaks down (e.g., rare but important features).

	9. Flattening discards spatial structure. Either narrow the claim to tabular data or provide a conv-friendly variant (patch- or channel-wise PCA seeding) and results that demonstrate actual vision benefits.

	10. What is the overhead of computing PCs relative to training time? Please release code, add a training-details table (data prep, hyperparams, hardware), and move key quantitative results from the appendix into the main paper.

---

> ### Author Response · Authors · 2025-11-21
> **Response to Reviewer GuXF**
>
> We sincerely thank you for your exceptionally thorough and constructive review.  We have performed a major revision to address your concerns:
>
>
> **On  Theoretical Inconsistencies and Reproducibility (Weakness i, vii, Questions 1, 2, 10):**
>
> To ensure scale invariance during the extraction of principal components, all input features were standardized prior to model initialization. Specifically, we applied normalization to the data, transforming features to have zero mean and unit variance on the training set. The same transformation parameters were applied to the test set.
>
> We have added an updated Appendix A.1 with more details on the hyperparameters and details on data preparation (we confirm **features are centered and scaled**), and have provided an **anonymized link to our code in the paper**. You can also access it at https://anonymous.4open.science/r/pcsinit-0675/README.md
>
> We have updated the **noise-handling and norm preservation** with the assumptions used and matrix shapes, and fixed related typos and notation issues. Though, the proofs do not rely on the normalization of the input, and therefore normalization is not included as an assumption. Also, the weight matrix $W_r$ becomes orthonormal automatically because of the mathematics of Principal Component Analysis (PCA). PCA uses Singular Value Decomposition (SVD) or eigenvalue decomposition, which by definition produces principal components (eigenvectors) that are orthogonal to each other and normalized to unit length.
>
> We have moved Figure 5, which demonstrates **running time** to the main paper, and analyzed it in the last paragraph of the Experiments section.

---

> ### Author Response · Authors · 2025-11-21
> **On the effects of hyperparameters (Weakness iv; Questions 3, 4, 8):**
>
> **$r$ was determined dynamically based on the explained variance ratio**. We selected the minimum number of components required to explain $95\%$ of the cumulative variance in the training data, as commonly used in many papers as cited in Appendix A.1. **The choice of $r$** affects the percentage of variance explained after PCA. *Therefore, increasing $r$ will likely increase the performance of the model to a certain threshold (for example, 95\% or 99\% of variance explained). However, a very high $r$ may lead to retaining noise, which may decrease the performance of the model, especially for noisy data.*
>
> Regarding the **bias and variance trade off**, for PCsInit and its variants, the number of retained principal components, **$r$, serves as a crucial structural hyperparameter that directly governs the model's bias-variance tradeoff**. When $r$ is set to a small value, the first layer acts as a narrow information bottleneck, aggressively filtering out input dimensions with lower eigenvalues. This regime typically results in high bias and low variance: the model is stable and robust to noise because it ignores the idiosyncratic fluctuations of the data, but it risks underfitting if the target variable relies on subtle, fine-grained features that were discarded during the dimension reduction.
>
> Conversely, selecting a large value for $r$ (approaching the total number of features) preserves nearly all the information present in the input, shifting the model towards a low bias and high variance state. In this regime, the initialization captures complex relationships and fine details, minimizing systematic error. However, because the lower-variance components often contain a lower Signal-to-Noise Ratio, a large $r$ allows the network to ingest and learn from random noise patterns, making the model more susceptible to overfitting and less consistent when generalizing to unseen data.
>
> **When the first layer is frozen throughout training**, PCsInit is equivalent to PCA-NN in terms of accuracy. Therefore, *the case where the first layer is frozen in PCsInit is also PCA-NN in the experiments*. Based on that, we have clarified in the revised manuscript that for** noisy dataset, the fact that "the PCsInit family generally performed better than PCA-NN" illustrates the benefits of fine-tuning**.
>
> When the first layer of PCsInit is not frozen completely,  the weights of the PCA-initialized layer are first frozen for some epochs. This stabilization phase allows the subsequent layers to adapt to the fixed principal component features. After that, the first layer was unfrozen, and the entire network was fine-tuned for the remainder of the training duration. Therefore, even though the paper chooses the default $n_{frozen} = 30$ epochs, **it is also reasonable to leave the first layers frozen, and then train the other layers until the loss saturates; after that, unfreeze the first layer and fine-tune the whole network, as in a standard transfer learning process**.
>
> **The model is sensitive to the first-layer freeze duration, and this represents a critical trade-off between preserving the robust, noise-filtering structure of the principal components and allowing the network to adapt to the specific supervised task**. If the freeze duration is too short, the immediate backpropagation of large error gradients from the randomly initialized upper layers risks destroying the orthonormal PCA subspace before the network can utilize its stability, effectively negating the intended noise-reduction benefits. Conversely, extending the freeze duration indefinitely constrains the model to the static PCA subspace, effectively reverting its performance to that of a standard PCA-NN and preventing the first layer from refining its features to capture discriminative signals that may not align with maximum variance. Consequently, the results are sensitive to this parameter because it defines the necessary transition point where the model shifts from a rigid, unsupervised prior to a flexible, task-optimized feature extractor.
>
> For **PCsInit-Sub, subsets are generated by randomly sampling a fixed percentage** (e.g., 20%) of the training instances to compute the initial principal components. This significantly reduces the computational overhead of the initial eigen decomposition while maintaining performance comparable to the full-data initialization, as indicated in the experiment results. Unlike streaming or incremental PCA, which continuously updates the covariance model to accommodate new data, PCsInit-Sub use one-time approximation and therefore, is likely to be less computationally expensive. However, this subset-based approach risks "breaking down" in scenarios where critical features are rare or sparse, as a random sample may fail to capture the variance of these infrequent but discriminative signals, which are inherently discarded or undersampled compared to high-variance global structures.

---

> ### Author Response · Authors · 2025-11-21
> **On Experimental Rigor & Baselines (Weakness ii, iii; Questions 5, 6):**
>
> - We have now included the strong baselines you recommended: a **standard MLP on raw (standardized) data** and a **ZCA whitening** pipeline.
>
> - The proposed techniques are for initializing the first layer of a network, and can be used to initialize the first layer of an autoencoder. Therefore, we have not included autoencoder in the comparison. However, thanks to your mentioning of autoencoder, we realize that PCsInit holds significant potential as a strategic initialization for the first layer of autoencoder architectures, particularly in accelerating the learning of compressed representations. By initializing the encoder's entry layer with the principal components, the model effectively begins optimization from a subspace that already captures the data's maximal variance, thereby "warm-starting" the reconstruction task compared to standard random initialization schemes. This approach theoretically aligns the network's initial trajectory with the optimal linear solution, allowing subsequent non-linear layers and fine-tuning epochs to focus immediately on capturing complex, non-linear residuals and fine-grained features rather than expending computational resources to re-learn basic structural components. Consequently, incorporating PCsInit into autoencoders could lead to faster convergence rates and enhanced reconstruction fidelity, while maintaining the noise-filtering benefits inherent to the PCA-based initialization. This would be a good direction for future work. Therefore, we included this as future works in the conclusion section of the revised manuscript.
>
> - We have also updated **all performance graphs** in the Appendix to include the **mean and 95% confidence intervals** from 10 runs, as you suggested.
>
> **On Interpretability (Weakness vi, Question 7):** We agree that our original claim was supported only by qualitative visuals. To address this, **we have added a new experiment with quantitative XAI metrics**, as you suggested. Specifically:
>
> - **Faithfulness:** Faithfulness is evaluated via the **Area Over the Perturbation Curve (AOPC)**, computed by zero-perturbation of features in Most Relevant First (MoRF) order. Figure 4 presents the AOPC distributions obtained by removing the top 1 to 5 Most Relevant First (MoRF) features, ordered by their absolute SHAP values, using zero perturbation across 10 runs on the Heart dataset. Overall, while both PCsInit and Raw MLP yield positive AOPC scores for class 0, their behaviors diverge for class 1: PCsInit remains positive, whereas NN becomes negative. This indicates that the top-ranked features consistently play a meaningful role in the PCsInit’s predictions for both classes.
>
> - **Stability:** It is quantified using the variance of the explanation values across runs, with lower variance indicating greater stability. Across 10 runs with different random seeds, SHAP achieves mean variance scores of 0.622 and 1.130 under PCsInit and Raw MLP, respectively, highlighting the higher stability of PCsInit explanations.
>
> - **Fidelity:** To evaluate fidelity, we generate five synthetic binary classification datasets following the *ssin* synthetic design used for classification in reference (4), each containing 2000 samples and 20 features. The number of predefined important features that directly influence the target ranges from 3 to 7 across the five datasets, while the remaining features are irrelevant or noisy (i.e., the first dataset has 3 important features, the second has 4, and so on). We then compare the top important features identified by SHAP under the PCsInit and Raw MLP models with the ground-truth important features. The mean recovery scores of both methods are above 80\%, demonstrating high fidelity in capturing the underlying ground-truth feature structure.
>
> References:
>
> (1) Evaluating the visualization of what a Deep Neural Network has learned https://arxiv.org/abs/1509.06321
>
> (2) Explainable artificial intelligence: A survey of needs, techniques, applications, and future direction https://arxiv.org/abs/2409.00265
>
> (3) Assessing Fidelity in XAI post-hoc techniques: A Comparative Study with Ground Truth Explanations Datasets https://arxiv.org/pdf/2311.01961
>
> (4) Using Sensitivity Analysis and Visualization Techniques to Open Black Box Data Mining Models https://scispace.com/pdf/using-sensitivity-analysis-and-visualization-techniques-to-2xqx36gbu2.pdf

---

> ### Author Response · Authors · 2025-11-21
> **On Scope and Vision Claims (Weakness v, Question 9):**
>
> In the setting of the paper, we follow a *flatten-then-PCA scheme* for image data such as  MNIST/CIFAR10, and since PCA discard the spatial structure, we have narrow down the claim for tabular data in the revised manuscript. Specifically, at the beginning of the "Methodologies", we have added: *"We assume $X$ represents tabular or vector data. For high-dimensional inputs with spatial or temporal structure (e.g., images), we assume a flattened vector representation, as PCsInit is inherently designed for fully connected architectures rather than convolutional ones."*
>
> In appendix A.1, we also updated the details to reflect this *"For the MNIST and Cifar10 datasets, the images are flattened before further processing,..."*
>
> We believe these substantial revisions, guided by your feedback, have significantly strengthened the paper's contributions, rigor, and theoretical correctness.  We kindly ask that you reconsider your score in light of these improvements.

---

> > ### Author Response · Authors · 2025-11-26
> > **Response to Reviewer GuXF**
> >
> > As the discussion period is drawing to a close, we wanted to briefly check if our previous responses and the revised manuscript fully addressed your concerns. We are happy to provide any further clarifications if needed.
> > Thank you very much.

---

### Note · Authors · 2026-01-26

I have read and agree with the venue's withdrawal policy on behalf of myself and my co-authors.

---

### Meta-Review · Area_Chair_1KeE · 2026-01-07

**Summary:**

The paper proposes a theoretically interesting initialization method (PCsInit) and demonstrates its potential. The authors strengthened the submission during the rebuttal by adding synthetic experiments. However, the empirical evaluation remains limited in scope, which makes it difficult to assess the method's effectiveness on a broader scale, and the contribution appeared incremental.

The reviewers presented a mixed view of the submission. In particular, Reviewer q5ia and Reviewer GuXF raised significant concerns regarding the soundness and contribution of the work. The primary negative concerns focused on:

**Limited Empirical Validation:** The initial submission relied heavily on the Heart dataset, which critical reviewers found insufficient to support general claims about neural network initialization and explainability.

**Soundness of Explainability Claims:** Reviewer GuXF (Soundness: 1) and Reviewer q5ia (Soundness: 2) questioned whether the proposed PCsInit method truly guarantees better explainability, requiring more rigorous ground-truth validation.

**Presentation and Scope:** Reviewers noted that the presentation was "fair" (2) and that the contribution appeared incremental or not fully established against broader baselines.

**Reviewer Concerns:**

Despite the addition of synthetic data, the concern regarding the scale and generality of the evaluation remains. While the synthetic comparison to "Raw MLP" is helpful, Reviewer q5ia's concern suggests a need for comparison against other state-of-the-art initialization or XAI-specific architectures to prove distinct superiority.

**Reviewer Scores:**

Reviewer 5xdt, who gave the highest score, did not provide an informative review. Reviewer q5ia, who gave the most critical score, and Reviewer GuXF likely would have maintained the score, while Reviewer SGM9 indicated the willingness to raise the score from 4 to 6.

---

### Decision · Program_Chairs · 2026-01-26

Reject